# FSOD-VFM: Few-Shot Object Detection with Vision Foundation Models and Graph Diffusion

**Chen-Bin Feng**[1,2*], **Youyang Sha**[2*], **Longfei Liu**[2], **Yongjun Yu**[2], **Chi Man Vong**[1†],
**Xuanlong Yu** [2†], **Xi Shen**[2†]
[1]University of Macau    [2]Intellindust AI Lab

## Abstract

In this paper, we present FSOD-VFM: Few-Shot Object Detectors with Vision Foundation Models, a framework that leverages vision foundation models to tackle the challenge of few-shot object detection. FSOD-VFM integrates three key components: a universal proposal network (UPN) for category-agnostic bounding box generation, SAM2 for accurate mask extraction, and DINOv2 features for efficient adaptation to new object categories. Despite the strong generalization capabilities of foundation models, the bounding boxes generated by UPN often suffer from overfragmentation, covering only partial object regions and leading to numerous small, false-positive proposals rather than accurate, complete object detections. To address this issue, we introduce a novel graph-based confidence reweighting method. In our approach, predicted bounding boxes are modeled as nodes in a directed graph, with graph diffusion operations applied to propagate confidence scores across the network. This reweighting process refines the scores of proposals, assigning higher confidence to whole objects and lower confidence to local, fragmented parts. This strategy improves detection granularity and effectively reduces the occurrence of false-positive bounding box proposals. Through extensive experiments on Pascal-$5^i$, COCO-$20^i$, and CD-FSOD datasets, we demonstrate that our method substantially outperforms existing approaches, achieving superior performance without requiring additional training. Notably, on the challenging CD-FSOD dataset, which spans multiple datasets and domains, our FSOD-VFM achieves 31.6 AP in the 10-shot setting, substantially outperforming previous training-free methods that reach only 21.4 AP. Code is available at: `https://intellindust-ai-lab.github.io/projects/FSOD-VFM`.

## 1 Introduction

Few-shot object detection (FSOD) (Han et al., 2022a; Wang et al., 2020; Xiong, 2023), the ability to detect objects with very few labeled samples, is a critical task for real-world applications where labeled data are scarce or expensive to obtain. Precisely, in FSOD, only a small support set provides the limited labeled bounding boxes for each category, and the goal is to detect those categories in unseen images. For instance, in a K-shot setting, only K-annotated bounding boxes per class are provided. The demand for efficient and generalizable solutions in fields like autonomous driving, robotics, and medical imaging has rendered this task increasingly critical.

Recent years, vision foundation models (VFMs), which are pre-trained on large-scale datasets, have demonstrated remarkable success across a wide range of vision tasks (Radford et al., 2021; Oquab et al., 2023; Siméoni et al., 2025; Jiang et al., 2024b; Ravi et al., 2024). These models, mostly based on transformer archtecture (Vaswani et al., 2017; Dosovitskiy et al., 2020), have set new benchmarks in tasks like object detection (Liu et al., 2024; Siméoni et al., 2025), segmentation (Siméoni et al., 2025), and image classification (Radford et al., 2021; Siméoni et al., 2025). Their ability to generalize across diverse tasks makes them appealing for solving complex vision challenges, partic-

---

*Equal contribution. † Corresponding author.

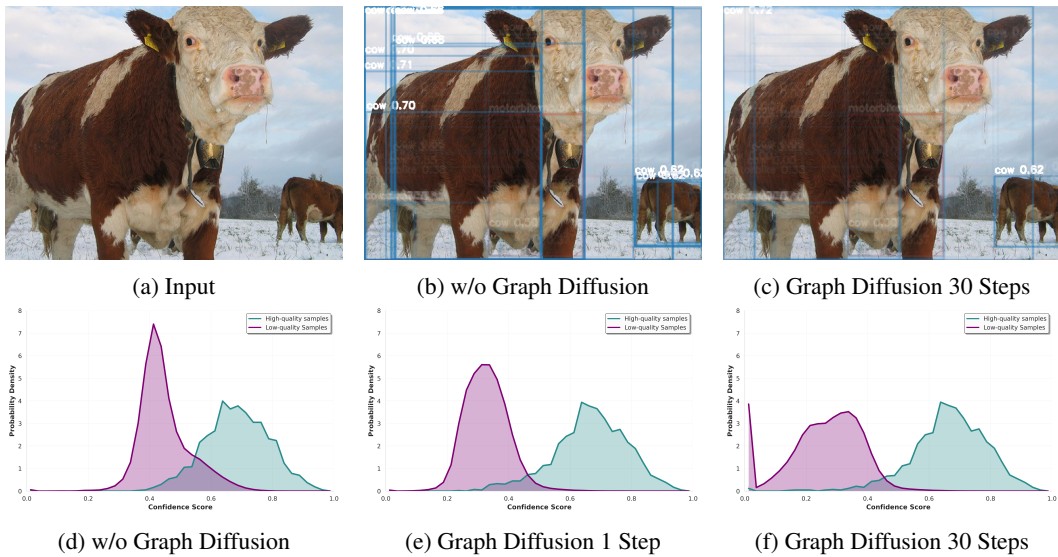

Figure 1: **Effect of graph diffusion.** The first row shows: (a) an input sample (Figure 1a), (b) its prediction without graph diffusion (Figure 1b), and (c) the final result after applying graph diffusion for 30 steps (Figure 1c). The second row illustrates how the distribution of high-quality boxes (IoU with any ground truth $> 0.75$) and low-quality boxes (IoU with any ground truth $< 0.1$) evolves: (d) without graph diffusion (Figure 1d), (e) after 1 step (Figure 1e), and (f) after 30 steps (Figure 1f).

ularly in settings that require minimal task-specific supervision. Beyond the benefits of fine-tuning for downstream tasks, vision foundation models also enable certain tasks to be performed in a fully unsupervised, zero-shot manner without any additional training. For example, DINO (Caron et al., 2021) can co-segment objects directly from extracted features (Amir et al., 2021; Wang et al., 2023). Likewise, combining SAM2 (Ravi et al., 2024) with a Kalman Filter yields a strong tracker that remains robust under occlusion (Yang et al., 2024), all without task-specific training.

An interesting research question arises: can powerful vision foundation models be directly applied to complex tasks such as object detection with only few-shot annotations, and even without additional training? In this work, we explore this question by leveraging foundation models to construct a universal few-shot object detector. Specifically, we build on three recent advances: a universal proposal network (UPN) (Jiang et al., 2024b) for category-agnostic bounding box generation, SAM2 (Ravi et al., 2024) for accurate mask extraction, and DINOv2 (Oquab et al., 2023) features for efficient adaptation to novel categories. However, directly applying this pipeline reveals a major limitation: bounding box overfragmentation. UPN often generates boxes that capture only small salient parts of objects rather than the full object, producing many redundant proposals for object fragments (See Figure 1b). This overfragmentation not only inflates false positives but also undermines detection accuracy and reliability. To overcome this issue, we propose a novel graph diffusion–based confidence refinement method that reweights bounding box proposals by exploiting their structural relationships, effectively suppressing local fragments and emphasizing complete objects.

Our graph diffusion approach represents predicted bounding boxes as nodes in a directed graph. Heat is propagated from one node to another when the latter not only receives a higher confidence score from UPN (Jiang et al., 2024b) but is also substantially overlapped by the former. By constructing this graph and performing diffusion over it, we propagate and refine decay scores across proposals. This formulation effectively captures relationships among candidate regions, promoting proposals that cover entire objects while suppressing fragmented parts. As a result, the detector assigns higher confidence to complete object regions and lower confidence to local parts, improving detection granularity. This effect is illustrated in Figure 1a, 1b, and 1c, where bounding box scores are visualized through transparency, showing how over-fragmented proposals are strongly suppressed after diffusion. A broader view on the first split of Pascal-5$^i$ (Everingham et al., 2010) under the 1-shot setting (Figures 1d, 1e, and 1f) further demonstrates that high-quality boxes (IoU with any ground truth $> 0.75$) retain their scores, while low-quality boxes (IoU with any ground truth $< 0.1$) are significantly

reduced. Extensive experiments on Pascal-$5^i$ (Everingham et al., 2010), COCO-$20^i$ (Kang et al., 2019; Lin et al., 2014), and CD-FSOD (Xiong, 2023) demonstrate that our method delivers significant improvements over previous approaches, enhancing performance without requiring additional training. Notably, on the challenging CD-FSOD dataset, which integrates six datasets from diverse domains, our FSOD-VFM achieves 31.6 AP in the 10-shot setting, substantially surpassing previous training-free methods that achieve only 21.4 AP.

In summary, our contributions are threefold:

- We present FSOD-VFM, a few-shot object detector that leverages vision foundation models to enable detection without any additional training.
- We design a graph diffusion–based confidence reweighting mechanism that effectively alleviates proposal over-fragmentation and enhances detection granularity.
- We conduct comprehensive experiments across multiple benchmarks, demonstrating that our method consistently and substantially outperforms existing approaches.

Our code is available at: `https://intellindust-ai-lab.github.io/projects/FSOD-VFM`.

## 2 RELATED WORK

**Few-shot object detection.** Few-shot object detection (FSOD) aims to detect unseen objects with limited labeled examples, addressing the challenge of generalizing to novel categories with only a few training samples. Transfer-learning methods (Chen et al., 2018; Wang et al., 2020; Qiao et al., 2021; Sun et al., 2021) leverage a model pre-trained on abundant base classes and adapt it to novel categories using fine-tuning schemes. Specifically, TFA (Wang et al., 2020) illustrates that fine-tuning only the final layers of a pre-trained model can effectively harness the capabilities of existing object detectors with minimal modifications. Meta-learning-based approach also plays a central role in FSOD. Meta R-CNN (Yan et al., 2019) incorporated meta-learning into two-stage detectors by designing class-aware attention mechanisms and dynamically reweighting features for novel categories. FSDetView (Xiao et al., 2022) introduces a feature aggregation to exploit the rich feature information shared from base classes. DE-ViT (Zhang et al., 2023) classifies proposals using similarity maps with support prototypes, rather than raw visual features, to improve generalization to novel classes. FM-FSOD (Han & Lim, 2024) achieves strong performance in few-shot object detection by leveraging a foundation model together with a large language model (LLM). Yet, all aforementioned methods require training. More recently, No-time-to-train (Espinosa et al., 2025) leverages a training-free strategy based on vision foundation models, demonstrating superior performance compared to training-based methods. In this work, we further explore training-free FSOD by integrating multiple vision foundation models and developing more effective ways of leveraging them, and pushing the frontier of this domain.

**Vision foundation models.** Vision foundation models have rapidly evolved and reshaped the downstream vision tasks. Thanks to the DINO series (Caron et al., 2021; Oquab et al., 2023; Siméoni et al., 2025), large-scale self-supervised learning has endowed the vision transformers with strong representation capabilities, enabling superior performance on tasks such as image classification and salient object segmentation with minimal adaptation. Furthermore, task-oriented foundation models have also emerged. Segment Anything Model (SAM) (Kirillov et al., 2023) and SAMv2 (Ravi et al., 2024) introduce a universal segmentation paradigm through interactive visual prompting and generate class-agnostic segmentation masks for the prompted areas. On the other hand, detection-oriented models such as the Universal Proposal Network (UPN) (Jiang et al., 2024b), which is based on detection transformers (Carion et al., 2020; Jiang et al., 2024a), can provide class-agnostic bounding boxes for the given image in coarse or fine-grained objectness modes. By unifying the complementary powers of diverse vision foundation models, our approach pushes the boundaries of zero-shot and few-shot object detection, delivering state-of-the-art performance without a single extra training step.

**Training-free applications with vision foundation models.** Beyond transfer learning, vision foundation models also enable powerful training-free applications. DINO (Caron et al., 2021) re-

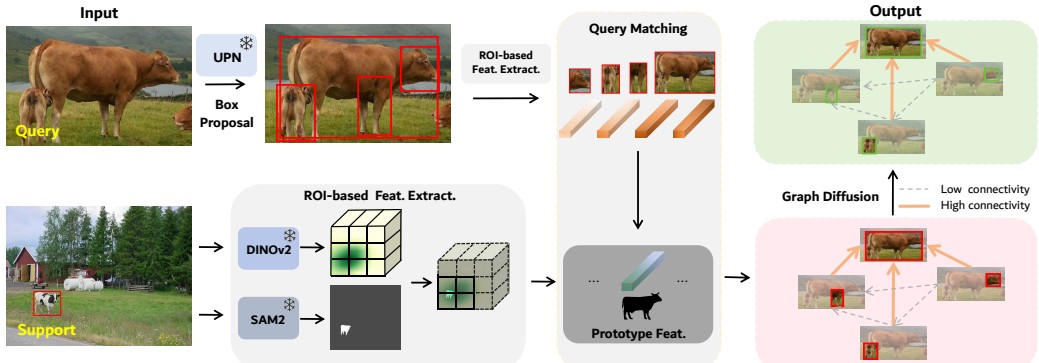

Figure 2: **Overview of FSOD-VFM.** Our method integrates UPN, SAM2, and DINOv2 to generate bounding box proposals and perform query matching. We build a graph and perform graph diffusion to mitigate over-fragmentation. The over-fragmented box regions appear more transparent after graph diffusion, indicating that their confidence has decayed.

veals object structures directly from attention maps, while LOST (Siméoni et al., 2021) and Token-Cut (Wang et al., 2023) enhance unsupervised object discovery, even in videos. Parts can likewise be segmented without supervision (Amir et al., 2021), and SAM2 (Ravi et al., 2024) combined with a Kalman Filter yields robust zero-shot tracking (Yang et al., 2024). These studies show that foundation models can solve complex tasks without task-specific training. Motivated by this, we design a training-free few-shot object detector. While concurrent work (Espinosa et al., 2025) also explores this direction, we demonstrate that stronger graph-based refinement yields substantially better performance.

## 3 METHOD

We propose FSOD-VFM, a framework designed for few-shot object detection. The method builds on three complementary vision foundation models: the Universal Proposal Network (UPN) (Jiang et al., 2024b) to generate category-agnostic bounding boxes, SAM2 (Ravi et al., 2024) to extract accurate masks based on these boxes, and DINOv2 (Oquab et al., 2023) as the feature extractor to obtain rich representations for both support and query objects. In the K-shot setting, we first denote the support set which contains K bounding box annotations:

$$\mathcal{S} = \{s^i\}_{i=1}^K, \quad s^i = (I^i, x_1^i, y_1^i, x_2^i, y_2^i, c^i)$$

where $(x_1^i, y_1^i, x_2^i, y_2^i)$ specifies the bounding box coordinates in image $I^i$, and $c^i \in \{1, \ldots, C\}$ denotes its class label among $C$ categories.

An overview of the FSOD-VFM pipeline is shown in Figure 2. For each annotation $s^i$ in the support set $\mathcal{S}$, we extract feature representations for the exact object region using SAM2 and DINOv2 according to the provided ground truth bounding boxes, and aggregate these features according to their category labels to form category-level prototypes. For each query image, the UPN generates bounding box region proposals, from which features are extracted and matched against the prototypes to produce class predictions and the corresponding matching scores. Finally, we propose to apply a graph diffusion mechanism to refine the scores of the bounding boxes, which helps suppress over-fragmented bounding boxes.

The section is organized as follows: Section 3.1 introduces the baseline, which leverages UPN (Jiang et al., 2024b), SAM2 (Ravi et al., 2024), and DINOv2 (Oquab et al., 2023) to construct category-level matching and perform bounding box classification. Section 3.2 details the graph diffusion procedure as well as the implementation details. Detailed pseudo-code for our FSOD-VFM method is provided in Appendix Section A.1.

## 3.1 Building few-shot object detectors with foundation models

**ROI-based feature extraction.** For a given annotation $s^i$, we perform Region-of-Interest (RoI) based feature extraction to obtain object-specific representations. Specifically, given a bounding box in $s^i$, we first generate a binary object mask $M^i$ in image $I^i$ using SAM2 (Ravi et al., 2024). Next, we extract a dense feature map from the whole image $I^i \in \mathbb{R}^{3 \times H \times W}$ with DINOv2 (Oquab et al., 2023) to obtain $F^i_{\text{img}} \in \mathbb{R}^{C \times H' \times W'}$, where we do additional downsampling from the original size. The bounding box is then mapped to the coordinates of the feature map. By interpolating $M^i$ to the feature map resolution, we obtain $M^i_{down}$, which allows us to pool features focusing only on the object region inside the bounding box:

$$F^i_s = \frac{1}{N_{\text{mask}}} \sum_{u=y'_1}^{y'_2} \sum_{v=x'_1}^{x'_2} F^i_{\text{img}}[:, u, v]\, M^i_{down}[u, v], \quad N_{\text{mask}} = \sum_{u=y'_1}^{y'_2} \sum_{v=x'_1}^{x'_2} M^i_{down}[u, v]. \tag{1}$$

**Building prototype features.** After obtaining RoI-based features $\{F^i_s\}$ in i-th annotation from the support set, we aggregate them to form class-level prototypes. Specifically, for each class $c \in \{1, \ldots, C\}$, we collect all support features that belong to class $c$: $\mathcal{F}_c = \{F^i_s \mid c^i = c\}$. The prototype of class $c$ is then computed as the mean feature $p_c = \frac{1}{|\mathcal{F}_c|} \sum_{f \in \mathcal{F}_c} f$, followed by $\ell_2$-normalization to ensure unit length: $\hat{p}_c = \frac{p_c}{\|p_c\|_2}$. These normalized prototypes $\{\hat{p}_c\}$ serve as the category-level representations for matching query proposals.

**Matching query proposals.** For each query image, we first employ UPN (Jiang et al., 2024b) to generate bounding box proposals $(x^j_1, y^j_1, x^j_2, y^j_2)$ along with their corresponding class-agnostic object detection scores $s^j_{\text{upn}}$. We extract RoI-based features using SAM2 and DINOv2 for each proposal, yielding feature representation $F^j_q$ for proposal $j$. We can achieve the class prediction for each proposal $j$ by calculating the cosine similarity between the proposal feature and the category-level support prototypes $\{\hat{p}_c\}$:

$$\hat{c}^j = \arg\max_c \ \cos(F^j_q, \hat{p}_c) \tag{2}$$

Finally, for the j-th proposal, we record its feature $F^j_q$, predicted class $\hat{c}^j$, class-agnostic detection score $s^j_{\text{upn}}$, and correponding mask $M^j$. These quadruples $(F^j_q, \hat{c}^j, s^j_{\text{upn}}, M^J)$ serve as inputs for the subsequent graph diffusion step, which refines the bounding box scores and reduces over-fragmentation.

## 3.2 Graph Diffusion

While UPN is effective at generating bounding box proposals, it may produce fragmented detections for a single object. To address this issue, we adopt a graph-based approach.

**Construction of the graph.** We construct a directed graph $\mathcal{G} = (\mathcal{V}, \mathcal{E})$, where $\mathcal{V}$ and $\mathcal{E}$ denote the sets of vertices and edges, respectively, and the graph has $N$ nodes, i.e., $|\mathcal{V}| = N$, indicating $N$ proposals within the same class in the query image. The $j$-th node stores the quadruple $(F^j_q, \hat{c}^j, s^j_{\text{upn}}, M^j)$. The edge connecting from i to j is defined as follows:

$$\mathcal{E}^{i,j} = \begin{cases} 0, & \text{if } s^i_{\text{upn}} > s^j_{\text{upn}}, \\ \dfrac{\text{Area}(M^i \cap M^j)}{\text{Area}(M^i)}, & \text{otherwise.} \end{cases} \tag{3}$$

The edge $\mathcal{E}^{i,j}$ characterizes the diffusion of energy from node $i$ to node $j$. Nodes with higher UPN scores are treated as high-quality proposals and thus retain their energy without diffusion. In contrast, nodes with lower UPN scores diffuse their energy toward more confident nodes. The diffusion factor is determined by the spatial coverage of node $j$ over node $i$, following the idea of suppressing fragmented proposals while consolidating coherent object regions, inspired by Espinosa et al. (2025).

| Method | F.T. Novel. | Novel Split 1 | | | | | Novel Split 2 | | | | | Novel Split 3 | | | | | Avg |
|---|---|---|---|---|---|---|---|---|---|---|---|---|---|---|---|---|---|
| | | 1 | 2 | 3 | 5 | 10 | 1 | 2 | 3 | 5 | 10 | 1 | 2 | 3 | 5 | 10 | |
| FsDetView (Xiao et al., 2022) | ✓ | 25.4 | 20.4 | 37.4 | 36.1 | 42.3 | 22.9 | 21.7 | 22.6 | 25.6 | 29.2 | 32.4 | 19.0 | 29.8 | 33.2 | 39.8 | 29.2 |
| TFA (Wang et al., 2020) | ✓ | 39.8 | 36.1 | 44.7 | 55.7 | 56.0 | 23.5 | 26.9 | 34.1 | 35.1 | 39.1 | 30.8 | 34.8 | 42.8 | 49.5 | 49.8 | 39.9 |
| Retentive RCNN (Fan et al., 2021) | ✓ | 42.4 | 45.8 | 45.9 | 53.7 | 56.1 | 21.7 | 27.8 | 35.2 | 37.0 | 40.3 | 30.2 | 37.6 | 43.0 | 49.7 | 50.1 | 41.1 |
| DiGeo (Ma et al., 2023) | ✓ | 37.9 | 39.4 | 48.5 | 58.6 | 61.5 | 26.6 | 28.9 | 41.9 | 42.1 | 49.1 | 30.4 | 40.1 | 46.9 | 52.7 | 54.7 | 44.0 |
| HeteroGraph (Han et al., 2021) | ✓ | 42.4 | 51.9 | 55.7 | 62.6 | 63.4 | 25.9 | 37.8 | 46.6 | 48.9 | 51.1 | 35.2 | 42.9 | 47.8 | 54.8 | 53.5 | 48.0 |
| Meta Faster R-CNN (Han et al., 2022a) | ✓ | 43.0 | 54.5 | 60.6 | 66.1 | 65.4 | 27.7 | 35.5 | 46.1 | 47.8 | 51.4 | 40.6 | 46.4 | 53.4 | 59.9 | 58.6 | 50.5 |
| CrossTransformer (Han et al., 2022b) | ✓ | 49.9 | 57.1 | 57.9 | 63.2 | 67.1 | 27.6 | 34.5 | 43.7 | 49.2 | 51.2 | 39.5 | 54.7 | 52.3 | 57.0 | 58.7 | 50.9 |
| LVC (Kaul et al., 2022) | ✓ | 54.5 | 53.2 | 58.8 | 63.2 | 65.7 | 32.8 | 29.2 | 50.7 | 49.8 | 50.6 | 48.4 | 52.7 | 55.0 | 59.6 | 59.6 | 52.3 |
| NIFF (Guirguis et al., 2023) | ✓ | 62.8 | 67.2 | 68.0 | 70.3 | 68.8 | 38.4 | 42.9 | 54.0 | 56.4 | 54.0 | 56.4 | 62.1 | 61.2 | 64.1 | 63.9 | 59.4 |
| Multi-Relation Det (Fan et al., 2020a) | ✗ | 37.8 | 43.6 | 51.6 | 56.5 | 58.6 | 22.5 | 30.6 | 40.7 | 43.1 | 47.6 | 31.0 | 37.9 | 43.7 | 51.3 | 49.8 | 43.1 |
| DE-ViT (ViT-S/14) (Zhang et al., 2023) | ✗ | 47.5 | 64.5 | 57.0 | 68.5 | 67.3 | 43.1 | 34.1 | 49.7 | 56.7 | 60.8 | 52.5 | 62.1 | 60.7 | 61.4 | 64.5 | 56.7 |
| DE-ViT (ViT-B/14) (Zhang et al., 2023) | ✗ | 56.9 | 61.8 | 68.0 | 73.9 | 72.8 | 45.3 | 47.3 | 58.2 | 59.8 | 60.6 | 58.6 | 62.3 | 62.7 | 64.6 | 67.8 | 61.4 |
| DE-ViT (ViT-L/14) (Zhang et al., 2023) | ✗ | 55.4 | 56.1 | 68.1 | 70.9 | 71.9 | 43.0 | 39.3 | 58.1 | 61.6 | 63.1 | 58.2 | 64.0 | 61.3 | 64.2 | 67.3 | 60.2 |
| No-Time-To-Train (Espinosa et al., 2025) | ✗ | 70.8 | 72.3 | 73.3 | 77.2 | 79.1 | 54.5 | 67.0 | 76.3 | 75.9 | 78.2 | 61.1 | 67.9 | 71.3 | 70.8 | 72.6 | 71.2 |
| **FSOD-VFM** | ✗ | **77.5** | **82.3** | **83.0** | **85.8** | **85.8** | **64.8** | **68.0** | **77.4** | **79.5** | **81.6** | **65.3** | **75.1** | **78.7** | **78.2** | **79.3** | **77.5** |

Results for competing methods are taken from Zhang et al. (2023), with the best highlighted in **bold**.

Table 1: **Results on Pascal-$5^i$ (Everingham et al., 2010).** We report nAP50, i.e., the average precision at IoU 0.5 on novel classes.

**Graph diffusion.** The diffusion process follows a scheme inspired by the classic PageRank algorithm (Brin & Page, 1998). Each node is assigned a prior weight $\mathbf{w}$, where

$$\mathbf{w}^i = \max_j \left( \mathcal{E}^{i,j} \right), \tag{4}$$

which reflects the strongest outgoing relation of node $i$ to other nodes. The initial state is set to $\boldsymbol{\pi}^0 = \left[ \frac{1}{N}, \frac{1}{N}, \ldots, \frac{1}{N} \right]^T$, and the transition matrix $\mathbf{P}$ is obtained by row-normalizing $\mathcal{E}$. The iterative diffusion is then defined as

$$\boldsymbol{\pi}^{t+1} = \alpha \mathbf{P} \otimes \boldsymbol{\pi}^t + (1 - \alpha)\mathbf{w}, \tag{5}$$

where $\alpha$ is the restart probability that balances global exploration with personalized propagation, and $\otimes$ denotes point-wise multiplication. To improve efficiency, we employ early stopping when $\|\boldsymbol{\pi}^{t+1} - \boldsymbol{\pi}^t\| < \tau$.

After convergence, the stationary distribution $\hat{\boldsymbol{\pi}}$ serves as a confidence weighting vector. For our task, what we ultimately need is a measure that penalizes low-score nodes. Therefore, after obtaining the stationary distribution $\hat{\boldsymbol{\pi}}$, we apply a simple transformation, which consists of taking its negative and adding 1, to convert it into a penalty score. The score of the $j$-th proposal is defined as

$$\hat{f}^j = (1 - \hat{\boldsymbol{\pi}}_j)^\lambda \max_c \cos(F_q^j, \hat{p}_c), \tag{6}$$

where proposals with high diffusion scores are downweighted, while confident and coherent proposals are preserved. The $\lambda$ is a hyperparameter that controls the strength of decay. The final prediction for the $j$-th proposal consists of the class $\hat{c}^j$ in Equation 2 and the score $\hat{f}^j$.

**Comparison between Graph Diffusion and NMS** Although both Graph Diffusion and Non-Maximum Suppression (NMS) reduce redundant proposals, their underlying principles differ fundamentally. NMS makes hard suppression decisions based solely on bounding box IoU and confidence scores. If two boxes overlap beyond a fixed threshold, the lower-score box is directly removed. This binary rule often suppresses valid detections when objects overlap or when bounding boxes are imperfect. Graph diffusion, in contrast, does not remove boxes based on thresholds. Instead, it refines proposal scores by propagating information between proposals using mask-level relationships from SAM2 (Ravi et al., 2024), which capture object boundaries much more accurately than bounding boxes. Through diffusion, unreliable proposals naturally receive lower scores, whereas reliable ones gain consistent support.

### 3.3 IMPLEMENTATION DETAILS.

We adopt SAM2-L (Hierarchical ViT) (Ravi et al., 2024) for mask extraction and DINOv2-L (Oquab et al., 2023) for feature extraction. During preprocessing, DINOv2 resizes all input images to 630 × 630, which is aligned with its pretrained patch size of 14 (since 630 is a multiple of 14). We additionally utilize DINOv3 (Siméoni et al., 2025), which accepts 624 × 624 input images, consistent with its patch size of 16. An ablation study on different feature extractors is presented in Section 4.2.

| Method | F.T. Novel. | 10-shot | | | 30-shot | | |
|---|---|---|---|---|---|---|---|
| | | nAP | nAP50 | nAP75 | nAP | nAP50 | nAP75 |
| TFA (Wang et al., 2020) | ✓ | 10.0 | 19.2 | 9.2 | 13.5 | 24.9 | 13.2 |
| FSCE (Sun et al., 2021) | ✓ | 11.9 | – | 10.5 | 16.4 | – | 16.2 |
| Retentive RCNN (Fan et al., 2021) | ✓ | 10.5 | 19.5 | 9.3 | 13.8 | 22.9 | 13.8 |
| HeteroGraph (Han et al., 2021) | ✓ | 11.6 | 23.9 | 9.8 | 16.5 | 31.9 | 15.5 |
| Meta F. R-CNN (Han et al., 2022a) | ✓ | 12.7 | 25.7 | 10.8 | 16.6 | 31.8 | 15.8 |
| LVC (Kaul et al., 2022) | ✓ | 19.0 | 34.1 | 19.0 | 26.8 | 45.8 | 27.5 |
| C. Transformer (Han et al., 2022b) | ✓ | 17.1 | 30.2 | 17.0 | 21.4 | 35.5 | 22.1 |
| NIFF (Guirguis et al., 2023) | ✓ | 18.8 | – | – | 20.9 | – | – |
| DiGeo (Ma et al., 2023) | ✓ | 10.3 | 18.7 | 9.9 | 14.2 | 26.2 | 14.8 |
| CD-ViTO (ViT-L) (Fu et al., 2024) | ✓ | 35.3 | 54.9 | 37.2 | 35.9 | 54.5 | 38.0 |
| FSRW (Kang et al., 2019) | ✗ | 5.6 | 12.3 | 4.6 | 9.1 | 19.0 | 7.6 |
| Meta R-CNN (Yan et al., 2019) | ✗ | 6.1 | 19.1 | 6.6 | 9.9 | 25.3 | 10.8 |
| DE-ViT (ViT-L) (Zhang et al., 2023) | ✗ | 34.0 | 53.0 | 37.0 | 34.0 | 52.9 | 37.2 |
| No-Time-To-Train (Espinosa et al., 2025) | ✗ | 36.6 | 54.1 | 38.3 | 36.8 | 54.5 | 38.7 |
| **FSOD-VFM** | ✗ | **44.0** | **59.4** | **47.6** | **45.8** | **61.9** | **49.4** |

Results for competing methods are taken from Fu et al. (2024), with the best highlighted in **bold**.

Table 2: **Results on COCO-20$^i$ (Kang et al., 2019; Lin et al., 2014).** We report nAP (IoU thresholds 0.5–0.95), nAP50 (IoU 0.5), and nAP75 (IoU threshold 0.75) on novel classes.

| Method | F.T Novel. | ArTaxOr | Clip art1k | DIOR | Deep Fish | NEU DET | UODD | Avg |
|---|---|---|---|---|---|---|---|---|
| TFA w/cos (Wang et al., 2020)° | ✓ | 3.1 / 8.8 / 14.8 | - | 8.0 / 18.1 / 20.5 | - | - | 4.4 / 8.7 / 11.8 | - |
| FSCE ○ (Sun et al., 2021)° | ✓ | 3.7 / 10.2 / 15.9 | - | 8.6 / 18.7 / 21.9 | - | - | 3.9 / 9.6 / 12.0 | - |
| DeFRCN (Qiao et al., 2021)° | ✓ | 3.6 / 9.9 / 15.5 | - | 9.3 / 18.9 / 22.9 | - | - | 4.5 / 9.9 / 12.1 | - |
| Distill-cdfsd (Xiong, 2023)° | ✓ | 5.1 / 12.5 / 18.1 | 7.6 / 23.3 / 27.3 | 10.5 / 19.1 / 26.5 | - / 15.5 / 15.5 | - / 16.0 / 21.1 | 5.9 / 12.2 / 14.5 | - / 16.4 / 20.5 |
| ViTDeT-FT† (Li et al., 2022)† | ✓ | 5.9 / 20.9 / 23.4 | 6.1 / 23.3 / 25.6 | 12.9 / 23.3 / 29.4 | 0.9 / 9.0 / 6.5 | 2.4 / 13.5 / 15.8 | 4.0 / 11.1 / 15.6 | 5.4 / 16.9 / 19.4 |
| Detic-FT (Zhou et al., 2022)† | ✓ | 3.2 / 8.7 / 12.0 | 15.1 / 20.2 / 22.3 | 4.1 / 12.1 / 15.4 | 9.0 / 14.3 / 17.9 | 3.8 / 14.1 / 16.8 | 4.2 / 10.4 / 14.4 | 6.6 / 13.3 / 16.5 |
| DE-ViT-FT (Zhang et al., 2023)† | ✓ | 10.5 / 38.0 / 49.2 | 13.0 / 38.1/40.8 | 14.7 / 23.4 / 25.6 | 19.3 / 21.2 / 21.3 | 0.6 / 7.8 / 8.8 | 2.4 / 5.0 / 5.4 | 10.1 / 22.3 / 25.2 |
| CD-ViTO (Fu et al., 2024)† | ✓ | 21.0 / 47.9 / 60.5 | 17.7 / 41.1 / 44.3 | 17.8 / 26.9 / 30.8 | 20.3 / 22.3 / 22.3 | 3.6 / 11.4 / 12.8 | 3.1 / 6.8 / 7.0 | 13.9 / 26.1 / 29.6 |
| Mixture (Liu et al., 2025) | ✓ | 26.1 / **63.3** / 71.3 | 20.1 / **45.1** / 49.9 | 20.6 / 32.1 / 37.8 | 24.2 / 29.5 / 34.1 | **9.1**/ 19.0 / 23.7 | 9.0 / 19.6 / **22.1** | 18.2 / **34.7** / 39.8 |
| Meta-RCNN (Yan et al., 2019)° | ✗ | 2.8 / 8.5 / 14.0 | - | 7.8 / 17.7 / 20.6 | - | - | 3.6 / 8.8 / 11.2 | - |
| Detic (Zhou et al., 2022)† | ✗ | 0.6 / 0.6 / 0.6 | 11.4 / 11.4 / 11.4 | 0.1 / 0.1 / 0.1 | 0.9 / 0.9 / 0.9 | 0.0 / 0.0 / 0.0 | 0.0 / 0.0 /0.0 | 2.2 / 2.2 / 2.2 |
| DE-ViT (Zhang et al., 2023)† | ✗ | 0.4 / 10.1 / 9.2 | 0.5 / 5.5 / 11.0 | 2.7 / 7.8 / 8.4 | 0.4 / 2.5 / 2.1 | 0.4 / 1.5 / 1.8 | 1.5 / 3.1 /3 .1 | 1.0 / 5.1 / 5.9 |
| No-Time-To-Train (Espinosa et al., 2025) | ✗ | 28.2 / 35.7 / 35.0 | 18.9 / 24.9 / 25.9 | 14.9 / 18.5 / 16.4 | 30.5 / 28.9 / 29.6 | 5.5 / 5.2 / 5.5 | 10.0 / **20.2** / 16.0 | 18.0 / 22.4 / 21.4 |
| **FSOD-VFM** | ✗ | **51.4** / 62.0 / 61.5 | **29.1** / 43.7 / 46.5 | 18.3 / 23.5 / 22.5 | **35.0 / 33.9 / 34.3** | 5.9 / 7.4 / 7.2 | **11.8** / 17.3 / 17.5 | **25.3** / 31.3 / 31.6 |

○ indicates Distill-CD-FSOD (Xiong, 2023) results, and † denotes CD-ViTO (Fu et al., 2024) results. Best results are in **bold**.

Table 3: **Results on the CD-FSOD benchmark (Xiong, 2023).** We report nAP (IoU thresholds 0.5–0.95) for all datasets, with each entry showing results for 1-shot, 5-shot, and 10-shot.

For proposals generated by UPN, we use "coarse" as the text prompt, discard bounding boxes with confidence scores below 0.01, and retain at most 500 proposals per image to accelerate inference. For the final predictions, we follow Espinosa et al. (2025) and output the top 100 bounding boxes per image ranked by confidence. In the graph diffusion process, we set $\alpha = 0.3$ and $\lambda = 0.5$, with a detailed study of these parameters provided in Section 4.2. We set the early stopping threshold $\tau$ to 1e-6. In terms of efficiency, inference on a single image takes approximately 2.4 seconds on a single NVIDIA A40 GPU without additional optimization.

# 4 EXPERIMENT

In this section, we provide experimental results. We evaluate our approach on Pascal-5$^i$ (Everingham et al., 2010), COCO-20$^i$ (Kang et al., 2019; Lin et al., 2014), and CD-FSOD (Xiong, 2023). For evaluation, we follow prior work (Xiong, 2023; Espinosa et al., 2025) and adopt the novel Average Precision (nAP) metric. This metric computes the mean Average Precision (mAP) over novel classes, which has become the standard evaluation protocol in few-shot object detection (Wang et al., 2020). The mAP is defined as the average precision across IoU thresholds from 0.5 to 0.95, following the COCO convention (Lin et al., 2014). For Pascal-5$^i$, we report nAP50, i.e., the AP at a single IoU threshold of 0.5 on novel categories, consistent with previous works (Wang et al., 2020; Han et al., 2022a; Xiao et al., 2022). Note that on Pascal-5$^i$ and COCO-20$^i$, most training-based approaches (Wang et al., 2020; Xiao et al., 2022) rely on training with both base classes and a few samples of novel classes. In contrast, our setup involves no additional training, making the task substantially more challenging. The description of datasets are detailed in Appendix A.3

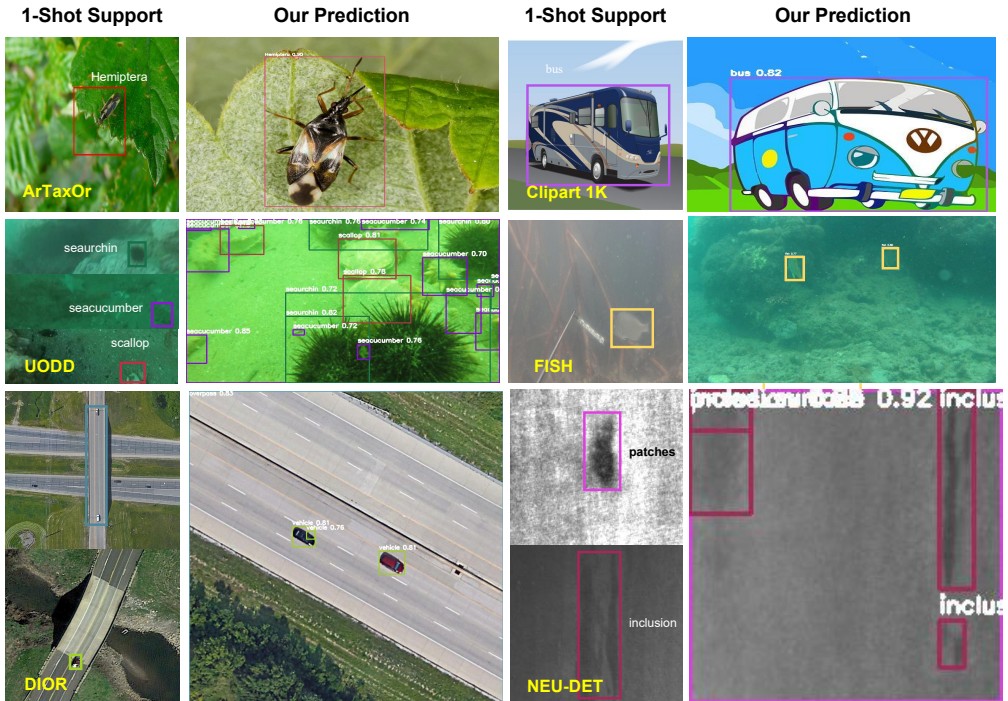

Figure 3: **Qualitative results of our method on CD-FSOD (Xiong, 2023) dataset.** Our method is shown to achieve precise object detection and classification across varied domains, with the sole requirement of one label per class.

## 4.1 COMPARISON TO STATE-OF-THE-ART METHODS

**Results on Pascal-5$^i$ (Everingham et al., 2010) and COCO-20$^i$ (Kang et al., 2019; Lin et al., 2014).** We present results on Pascal-5$^i$ and COCO-20$^i$ in Table 1 and Table 2, comparing with methods trained on novel classes and methods not trained on novel classes. Under the same training-free setting, our method delivers significant and consistent improvements over the recent concurrent work No-Time-To-Train (Espinosa et al., 2025). Notably, it also outperforms many training-based approaches by a large margin. These results demonstrate the effectiveness of the proposed FSOD-VFM, showing that with pre-trained large vision foundation models, carefully designed strategies can yield strong few-shot object detection performance.

**Results on CD-FSOD (Xiong, 2023).** We report experimental results on the more challenging CD-FSOD dataset in Table 3, where our method achieves consistent performance gains over all training-free baselines. Even in comparison with methods fine-tuned on novel categories, our approach remains highly competitive. This superiority is particularly pronounced in the one-shot setting. Our method delivers exceptional performance on ArTaxOr and Clipart1k, two datasets whose domains are close to natural images. In contrast, NEU-DET, Deep Fish, and UODD pose substantially greater challenges: these benchmarks involve detecting low-saliency defect objects in microscopic textures or blurred targets in underwater scenes. For such domain-specific scenarios, fine-tuned methods yield better performance than our training-free approach, which is an expected trade-off for not using target domain fine-tuning. Notably, Mixture (Liu et al., 2025) leverages DINO Detector (Zhang et al., 2022) and DINOv2 (Oquab et al., 2023), which are both pretrained on large-scale corpora and further fine-tuned on target domains. It achieves state-of-the-art performance on this dataset. Even so, our FSOD-VFM maintains a performance advantage over Mixture in the one-shot case, which underscores the strong robustness of our training-free framework. Qualitative visual examples in Figure 3 and segmentation examples in Figure 4 illustrate the inherent difficulty and rich domain diversity of the CD-FSOD benchmark.

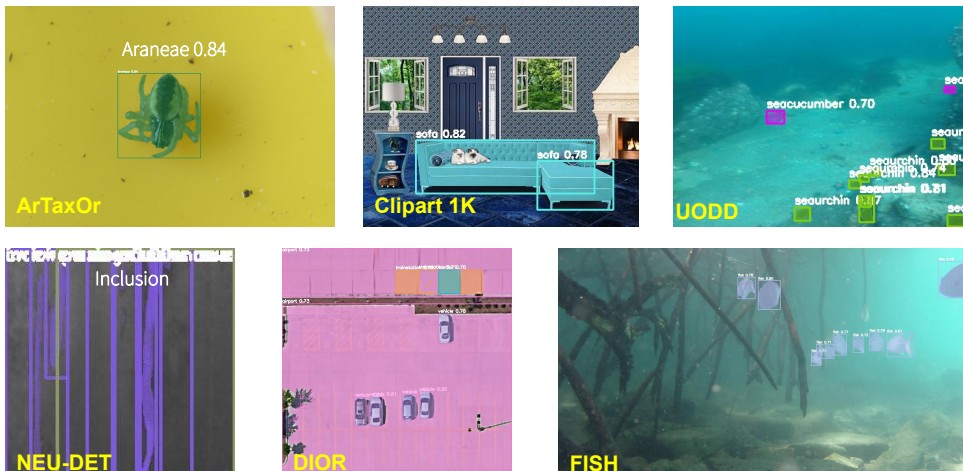

Figure 4: **Segmentation visualization of CD-FSOD (Xiong, 2023).** SAM2 delivers reliable segmentation results across most scenarios (e.g., insect-related, cartoon, and deep-sea scenes).

| Method | Pascal-5$^i$ | COCO-20$^i$ |
|---|---|---|
| w/o Refine. ($\lambda = 0$) | 7.4 | 9.9 |
| + NMS | 23.4 | 26.1 |
| + Soft NMS (Bodla et al., 2017) | 28.1 | 26.6 |
| + WBF (Solovyev et al., 2021) | 25.5 | 27.8 |
| + Soft Merging (Espinosa et al., 2025) | 66.0 | 50.4 |
| + **Graph Diffusion (Ours)** | **77.5** | **59.4** |
| + Graph Diffusion + NMS | 77.2 | 59.1 |

Table 4: **Post-processing comparison of nAP50 on Pascal-5$^i$ (1-shot) and COCO-20$^i$ (10-shot).**

| $\lambda$ in Eqn. 6 | $\alpha$ in Eqn. 5 | $t$ in Eqn. 5 | Pascal-5$^i$ | COCO-20$^i$ | Infer. Time (s / img) |
|---|---|---|---|---|---|
| 0.3 | 0.0 | 30 | 76.5 | 58.9 | 2.42 |
| 0.5 | 0.0 | 30 | 77.1 | 59.0 | 2.42 |
| 1.0 | 0.0 | 30 | 76.8 | 58.7 | 2.42 |
| 0.3 | 0.3 | 30 | 77.0 | 59.5 | 2.42 |
| **0.5** | **0.3** | **30** | **77.5** | 59.4 | 2.42 |
| 1.0 | 0.3 | 30 | 77.1 | 59.0 | 2.42 |
| 0.3 | 0.5 | 30 | 77.4 | 58.9 | 2.42 |
| 0.5 | 0.5 | 30 | 77.2 | **59.7** | 2.42 |
| 1.0 | 0.5 | 30 | 77.4 | 59.3 | 2.42 |
| 0.5 | 0.3 | 1 | 76.5 | 59.1 | 2.37 |
| 0.5 | 0.3 | 5 | 77.5 | 59.4 | 2.39 |
| 0.5 | 0.3 | 10 | 77.5 | 59.4 | 2.39 |
| **0.5** | **0.3** | **30** | **77.5** | **59.4** | 2.42 |
| 0.5 | 0.3 | 100 | 77.5 | 59.4 | 2.48 |

Table 5: **Impact of $\lambda$, $\alpha$ and t of nAP50 on Pascal-5$^i$ (1-shot) and COCO-20$^i$ (10-shot).**

## 4.2 DISCUSSION AND ANALYSIS.

In this section, we analyze the impact of different modules in FSOD-VFM. All experiments are conducted on Pascal-5$^i$ split1 under 1-shot setting and COCO-20$^i$ under the 10-shot setting.

**Graph diffusion versus other post-processing.** We compare the proposed graph diffusion to other post-processing approaches, which including: Non-Maximum Suppression (NMS), Soft NMS (Bodla et al., 2017), Weighted-Box-Fusion (WBF) (Solovyev et al., 2021), and Soft Merging (Espinosa et al., 2025). The results are provided in Table 4. We demonstrate that our graph diffusion approach achieves significantly better performance. In addition, our graph diffusion integrated with NMS delivers marginally inferior performance. This observation demonstrates that NMS is not necessary for the reweighting pipeline proposed in this study.

**Hyper-parameters in graph diffusion.** We analyze the impact of key hyperparameters in the graph diffusion process, including the decay factor $\lambda$ in Equation 6, the restart probability $\alpha$ in Equation 5, and the number of diffusion steps $t$ in Equation 5. The results are summarized in Table 5. From the table, we can observe that our method exhibits low sensitivity to these hyperparameters, resulting in relatively consistent performance across different settings. Among them, $\lambda = 0.5$, $\alpha = 0.3$ or $\alpha = 0.5$, $\alpha = 0.5$ yield the best performance, we choose $\lambda = 0.5$, $\alpha = 0.3$ for all datasets. Additionally, performance improves with increasing diffusion steps and stabilizes after the fifth steps, indicating approximate convergence. The diffusion process fully converge around 70 steps (illustrated in Appendix A.2). To balance between computation time and stable performance, we adopt 30 diffusion steps in all our experiments.

| DINO Backbone | v2-S | v2-S-Reg | v2-B | v2-B-Reg | v2-L | v2-L-Reg | v2-G | v2-G-Reg | v3-S | v3-Splus | v3-B | v3-L | v3-Hplus |
|---|---|---|---|---|---|---|---|---|---|---|---|---|---|
| **Pascal-5$^i$** | 73.1 | 66.7 | **81.2** | 69.0 | 77.5 | 73.5 | 69.2 | 64.9 | 64.4 | 71.8 | 75.0 | 71.6 | 58.7 |
| **COCO-20$^i$** | 52.5 | 47.2 | 57.6 | 53.3 | **59.4** | 56.4 | 58.4 | 53.3 | 48.3 | 52.2 | 55.5 | 56.2 | 45.5 |

Table 6: **Ablation results of nAP50 on Pascal-5$^i$ (1-shot) and COCO-20$^i$ (10-shot) on different backbone choices for FSOD-VFM.** '-Reg' denotes the backbone with register (Oquab et al., 2023; Darcet et al., 2023).

**Impact of feature extractor.** In the proposed FSOD-VFM, we adopt DINOv2-L without registration as our feature extractor. We conduct a full ablation study on different backbone choices, including various sizes of DINOv2 (Oquab et al., 2023), DINOv2 with registers (Darcet et al., 2023), and the more recent DINOv3 (Siméoni et al., 2025). The results are summarized in Table 6. The DINOv2-B achieve the best performance in Pascal-5$^i$ split1 (1-shot). However, its overall performance is weaker than DINOv2-L (illustrated in Appendix A.4).

**Limitations.** Despite the significant improvements achieved by FSOD-VFM, our approach has several limitations. First, the performance gain decreases as the number of annotated samples increases; for example, the improvement from 5-shot to 10-shot is marginal compared to that from 1-shot to 5-shot. Second, FSOD-VFM incurs a relatively high inference cost, as it relies on multiple foundation models and is not optimized for real-time applications. Nevertheless, the method requires very few annotations and no additional training, making it a powerful tool for data analysis and applicable across diverse domains.

## 5 CONCLUSION

This paper proposed FSOD-VFM, a few-shot object detection framework leveraging vision foundation models, integrating category-agnostic UPN for class-agnostic bounding box generation, SAM2 for mask extraction, and DINOv2 for feature extraction. To address bounding box overfragmentation in the prediction, characterized by partial object coverage and excessive false positives, we designed a graph-based confidence reweighting method. This method modeled predicted boxes as nodes in a directed graph, then employed graph diffusion to propagate confidence scores. This graph diffusion-based approach elevates scores for whole-object proposals while lowering those for fragmented local parts, thereby enhancing detection granularity and reducing false positives. Experiments on Pascal-5$^i$, COCO-20$^i$, and CD-FSOD showed that FSOD-VFM outperformed existing methods without extra training. We believe this work offers a novel perspective on leveraging foundation models to address visual tasks in a training-free paradigm.

**Acknowledgment** We thank Intellindust for their support, in particular Caizhi Zhu and Xiao Zhou. This work was supported in part by the Shenzhen Science and Technology Innovation Committee under Project SGDX20220530111001006, and in part by the Science and Development Fund of Macau under Grants 0118/2024/RIA2 and 0216/2024/AGJ.

## ETHICS STATEMENT

We declare that they do not find any potential violations of the ICLR code of ethics.

## REPRODUCIBILITY STATEMENT

Our implementation details are provided in Section 3.3, including all necessary hyperparameters. A general algorithm is presented in Appendix A.1. All datasets used in our work are publicly available: Pascal VOC can be accessed at `http://host.robots.ox.ac.uk/pascal/VOC`, COCO is available at `https://cocodataset.org`, and CD-FSOD can be downloaded from `https://yuqianfu.com/CDFSOD-benchmark`.

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

# A    APPENDIX

## A.1    PSEUDO-CODE OF FSOD-VFM

We present the pseudo code of FSOD-VFM in Algorithm 1.

---

**Algorithm 1** FSOD-VFM: Few-Shot Object Detection with Foundation Models

---

1: **Input:** Support set $\mathcal{S} = \{s^i\}_{i=1}^K$, query image $I_q$, max iterations max_iter, diffusion parameters $\alpha, \lambda$
2: **Output:** Refined prediction of proposals
3: **Stage 1: Build Class Prototypes**
4: **for** each support annotation $s^i \in \mathcal{S}$ **do**
5:      Generate object mask $M^i$ using SAM2
6:      Extract dense feature map $F_{\text{img}}^i$ using DINOv2
7:      Pool features over mask: $F_s^i \leftarrow$ masked RoI pooling (Eq. 1)
8: **end for**
9: **for** each class $c \in \{1, \dots, C\}$ **do**
10:      Collect support features: $\mathcal{F}_c = \{F_s^i \mid c^i = c\}$
11:      Compute prototype: $p_c = \frac{1}{|\mathcal{F}_c|} \sum_{f \in \mathcal{F}_c} f$
12:      Normalize prototype: $\hat{p}_c = p_c / \|p_c\|_2$
13: **end for**
14: **Stage 2: Query Proposal Matching**
15: Generate bounding box proposals $\{(x_1^j, y_1^j, x_2^j, y_2^j)\}$ using UPN
16: **for** each proposal $j$ **do**
17:      Extract RoI features $F_q^j$ using SAM2 + DINOv2
18:      Predict class: $\hat{c}^j = \arg\max_c \cos(F_q^j, \hat{p}_c)$ (Eq. 2)
19:      Store quadruple $(F_q^j, \hat{c}^j, s_{\text{upn}}^j, M^j)$
20: **end for**
21: **Stage 3: Graph Diffusion for Score Refinement**
22: Construct graph $\mathcal{G} = (\mathcal{V}, \mathcal{E})$ with nodes as proposals in each class $c \in \{1, \dots, C\}$
23: **for** each class $c \in \{1, \dots, C\}$ **do**
24:      **for** each edge $(i, j)$ **do**
25:           **if** $s_{\text{upn}}^i > s_{\text{upn}}^j$ **then**
26:                $\mathcal{E}^{i,j} \leftarrow 0$
27:           **else**
28:                $\mathcal{E}^{i,j} \leftarrow \frac{\text{Area}(M^i \cap M^j)}{\text{Area}(M^i)}$
29:           **end if**
30:      **end for**
31:      Row-normalize edges: $\mathbf{P} \leftarrow$ transition matrix from $\mathcal{E}$
32:      Initialize prior weights: $\mathbf{w}^i \leftarrow \max_j(\mathcal{E}^{i,j})$
33:      Initialize diffusion vector: $\boldsymbol{\pi}^0 \leftarrow [1/N, \dots, 1/N]^T$
34:      **for** $t = 0$ to max_iter **do**
35:           $\boldsymbol{\pi}^{t+1} \leftarrow \alpha \mathbf{P} \otimes \boldsymbol{\pi}^t + (1 - \alpha)\mathbf{w}$
36:           **if** $\|\boldsymbol{\pi}^{t+1} - \boldsymbol{\pi}^t\| < \tau$ **then**
37:                **break**                                      ▷ Convergence achieved
38:           **end if**
39:      **end for**
40: **end for**
41: **for** each proposal $j$ **do**
42:      Refine score: $\hat{f}^j \leftarrow (1 - \hat{\pi}_j)^\lambda \cdot \max_c \cos(F_q^j, \hat{p}_c)$ (Eq. 6)
43: **end for**
44: **return** final proposals $\{(\hat{c}^j, \hat{f}^j)\}$

---

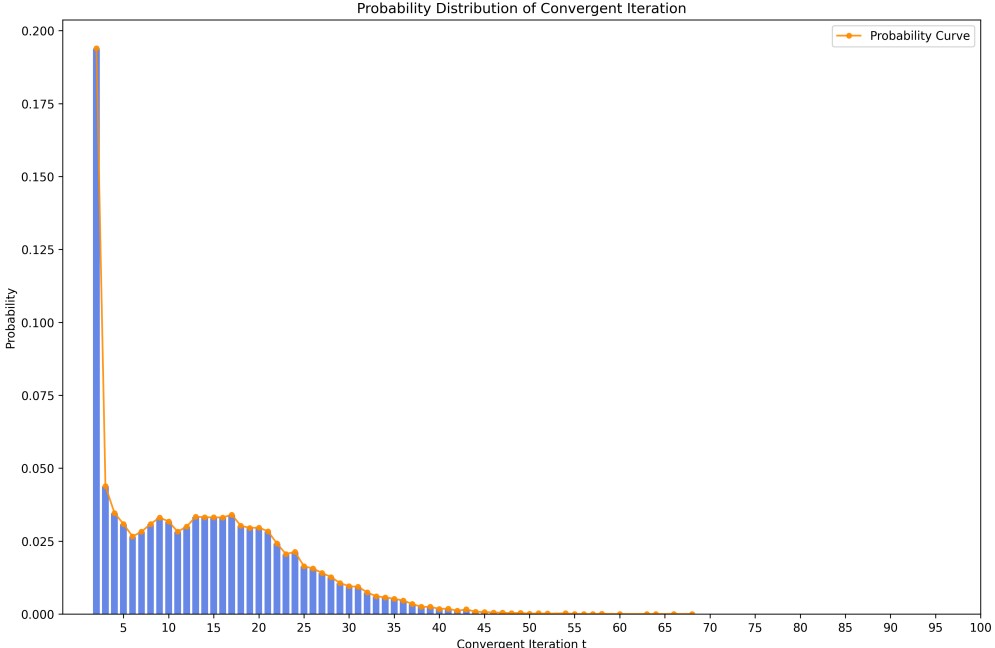

Figure 5: **Graph diffusion convergent iteration.** The graph diffusion process stop around 70 steps in Pascal-5$^i$ (Everingham et al., 2010) split1 under 1-shot setting. However, most runs end before 50 steps.

## A.2 DIFFUSION TIMESTEPS

The graph diffusion process converges in approximately 70 iterations, as illustrated in Figure 5. Most cases converge within 50 steps, with very few cases completing between 50-70 steps. However, as shown in Table 5, the AP performance stabilizes after 5 steps. This is because AP performance depends on the ordering of box proposals rather than the absolute values of confidence scores of each proposal. We opt to run 30 steps to balance stability and efficiency.

## A.3 DATASET DESCRIPTION

We describe the datasets used in this section.

**Pascal-5$^i$ (Everingham et al., 2010).** The Pascal VOC dataset (Everingham et al., 2010) contains 20 object categories. Following the standard few-shot protocol (Guirguis et al., 2023), it is divided into three splits, each with 5 novel classes and 15 base classes. Our setup performs no additional training and relies only on few-shot samples of novel categories, making the task more challenging. For Pascal-5$^i$, we follow prior works (Wang et al., 2020; Han et al., 2022a; Xiao et al., 2022) and report nAP50, i.e., the average precision on novel classes at an IoU threshold of 0.5. We use a random seed of 33 when sampling support sets, following the approach described in (Espinosa et al., 2025).

**COCO-20$^i$ (Kang et al., 2019; Lin et al., 2014).** The COCO dataset (Lin et al., 2014) contains 80 object categories, with 20 classes designated as novel following standard few-shot protocols (Wang et al., 2020). We report 10-shot and 30-shot results using the standard evaluation protocol. Similar to Pascal-5$^i$, our setup is training-free and relies solely on few-shot samples of novel categories, making the task more challenging. For COCO-20$^i$, we follow prior works (Wang et al., 2020; Han et al., 2022a; Xiao et al., 2022) and report nAP (IoU thresholds 0.5–0.95), nAP50 (IoU 0.5), and nAP75 (IoU 0.75) on novel classes.5) and nAP75 (IoU threshold 0.75) on novel classes. We use a random seed of 33 when sampling support sets, following the approach described in (Espinosa et al., 2025).

| Method | F.T. Novel. | Novel Split 1 | | | | | Novel Split 2 | | | | | Novel Split 3 | | | | | Avg |
|---|---|---|---|---|---|---|---|---|---|---|---|---|---|---|---|---|---|
| | | 1 | 2 | 3 | 5 | 10 | 1 | 2 | 3 | 5 | 10 | 1 | 2 | 3 | 5 | 10 | |
| FsDetView (Xiao et al., 2022) | ✓ | 25.4 | 20.4 | 37.4 | 36.1 | 42.3 | 22.9 | 21.7 | 22.6 | 25.6 | 29.2 | 32.4 | 19.0 | 29.8 | 33.2 | 39.8 | 29.2 |
| TFA (Wang et al., 2020) | ✓ | 39.8 | 36.1 | 44.7 | 55.7 | 56.0 | 23.5 | 26.9 | 34.1 | 35.1 | 39.1 | 30.8 | 34.8 | 42.8 | 49.5 | 49.8 | 39.9 |
| Retentive RCNN (Fan et al., 2021) | ✓ | 42.4 | 45.8 | 45.9 | 53.7 | 56.1 | 21.7 | 27.8 | 35.2 | 37.0 | 40.3 | 30.2 | 37.6 | 43.0 | 49.7 | 50.1 | 41.1 |
| DiGeo (Ma et al., 2023) | ✓ | 37.9 | 39.4 | 48.5 | 58.6 | 61.5 | 26.6 | 28.9 | 41.9 | 42.1 | 49.1 | 30.4 | 40.1 | 46.9 | 52.7 | 54.7 | 44.0 |
| HeteroGraph (Han et al., 2021) | ✓ | 42.4 | 51.9 | 55.7 | 62.6 | 63.4 | 25.9 | 37.8 | 46.6 | 48.9 | 51.1 | 35.2 | 42.9 | 47.8 | 54.8 | 53.5 | 48.0 |
| Meta Faster R-CNN (Han et al., 2022a) | ✓ | 43.0 | 54.5 | 60.6 | 66.1 | 65.4 | 27.7 | 35.5 | 46.1 | 47.8 | 51.4 | 40.6 | 46.4 | 53.4 | 59.9 | 58.6 | 50.5 |
| CrossTransformer (Han et al., 2022b) | ✓ | 49.9 | 57.1 | 57.9 | 63.2 | 67.1 | 27.6 | 34.5 | 43.7 | 49.2 | 51.2 | 39.5 | 54.7 | 52.3 | 57.0 | 58.7 | 50.9 |
| LVC (Kaul et al., 2022) | ✓ | 54.5 | 53.2 | 58.8 | 63.2 | 65.7 | 32.8 | 29.2 | 50.7 | 49.8 | 50.6 | 48.4 | 52.7 | 55.0 | 59.6 | 59.6 | 52.3 |
| NIFF (Guirguis et al., 2023) | ✓ | 62.8 | 67.2 | 68.0 | 70.3 | 68.8 | 38.4 | 42.9 | 54.0 | 56.4 | 54.0 | 56.4 | 62.1 | 61.2 | 64.1 | 63.9 | 59.4 |
| Multi-Relation Det (Fan et al., 2020b) | ✗ | 37.8 | 43.6 | 51.6 | 56.5 | 58.6 | 22.5 | 30.6 | 40.7 | 43.1 | 47.6 | 31.0 | 37.9 | 43.7 | 51.3 | 49.8 | 43.1 |
| DE-ViT (ViT-S/14) (Zhang et al., 2023) | ✗ | 47.5 | 64.5 | 57.0 | 68.5 | 67.3 | 43.1 | 34.1 | 49.7 | 56.7 | 60.8 | 52.5 | 62.1 | 60.7 | 61.4 | 64.5 | 56.7 |
| DE-ViT (ViT-B/14) (Zhang et al., 2023) | ✗ | 56.9 | 61.8 | 68.0 | 73.9 | 72.8 | 45.3 | 47.3 | 58.2 | 59.8 | 60.6 | 58.6 | 62.3 | 62.7 | 64.6 | 67.8 | 61.4 |
| DE-ViT (ViT-L/14) (Zhang et al., 2023) | ✗ | 55.4 | 56.1 | 68.1 | 70.9 | 71.9 | 43.0 | 39.3 | 58.1 | 61.6 | 63.1 | 58.2 | 64.0 | 61.3 | 64.2 | 67.3 | 60.2 |
| No-Time-To-Train (Espinosa et al., 2025) | ✗ | 70.8 | 72.3 | 73.3 | 77.2 | 79.1 | 54.5 | 67.0 | 76.3 | 75.9 | 78.2 | 61.1 | 67.9 | 71.3 | 70.8 | 72.6 | 71.2 |
| **FSOD-VFM-DINOv2-B** | ✗ | **81.2** | 82.1 | 82.4 | 85.2 | 84.9 | 62.9 | 68.4 | 75.8 | 75.8 | 81.0 | 61.6 | 73.1 | 76.8 | 75.5 | 76.8 | 76.2 |
| **FSOD-VFM-DINOv2-L** | ✗ | 77.5 | **82.3** | **83.0** | **85.8** | 85.8 | 64.8 | 68.0 | 77.4 | 79.5 | **81.6** | **65.3** | 75.1 | **78.7** | **78.2** | 79.3 | 77.5 |
| **FSOD-VFM-RADIOv4** | ✗ | 77.5 | 81.4 | 81.6 | 84.6 | **86.6** | **69.2** | **75.2** | **79.2** | **79.7** | 79.8 | 62.6 | **75.9** | 76.4 | 76.2 | **79.8** | **77.7** |

Results for competing methods are taken from Zhang et al. (2023), with the best highlighted in **bold**.

Table 7: **Results on Pascal-$5^i$ (Everingham et al., 2010).** We report nAP50, i.e., the average precision at IoU 0.5 on novel classes.

| Method | F.T. Novel. | 10-shot | | | 30-shot | | |
|---|---|---|---|---|---|---|---|
| | | nAP | nAP50 | nAP75 | nAP | nAP50 | nAP75 |
| TFA (Wang et al., 2020) | ✓ | 10.0 | 19.2 | 9.2 | 13.5 | 24.9 | 13.2 |
| FSCE (Sun et al., 2021) | ✓ | 11.9 | – | 10.5 | 16.4 | – | 16.2 |
| Retentive RCNN (Fan et al., 2021) | ✓ | 10.5 | 19.5 | 9.3 | 13.8 | 22.9 | 13.8 |
| HeteroGraph (Han et al., 2021) | ✓ | 11.6 | 23.9 | 9.8 | 16.5 | 31.9 | 15.5 |
| Meta F. R-CNN (Han et al., 2022a) | ✓ | 12.7 | 25.7 | 10.8 | 16.6 | 31.8 | 15.8 |
| LVC (Kaul et al., 2022) | ✓ | 19.0 | 34.1 | 19.0 | 26.8 | 45.8 | 27.5 |
| C. Transformer (Han et al., 2022b) | ✓ | 17.1 | 30.2 | 17.0 | 21.4 | 35.5 | 22.1 |
| NIFF (Guirguis et al., 2023) | ✓ | 18.8 | – | – | 20.9 | – | – |
| DiGeo (Ma et al., 2023) | ✓ | 10.3 | 18.7 | 9.9 | 14.2 | 26.2 | 14.8 |
| CD-ViTO (ViT-L) (Fu et al., 2024) | ✓ | 35.3 | 54.9 | 37.2 | 35.9 | 54.5 | 38.0 |
| FSRW (Kang et al., 2019) | ✗ | 5.6 | 12.3 | 4.6 | 9.1 | 19.0 | 7.6 |
| Meta R-CNN (Yan et al., 2019) | ✗ | 6.1 | 19.1 | 6.6 | 9.9 | 25.3 | 10.8 |
| DE-ViT (ViT-L) (Zhang et al., 2023) | ✗ | 34.0 | 53.0 | 37.0 | 34.0 | 52.9 | 37.2 |
| No-Time-To-Train (Espinosa et al., 2025) | ✗ | 36.6 | 54.1 | 38.3 | 36.8 | 54.5 | 38.7 |
| **FSOD-VFM-DINOv2-B** | ✗ | 42.4 | 57.6 | 45.8 | 43.7 | 59.6 | 47.1 |
| **FSOD-VFM-DINOv2-L** | ✗ | 44.0 | 59.4 | 47.6 | 45.8 | 61.9 | 49.4 |
| **FSOD-VFM-RADIOv4** | ✗ | **46.3** | **62.5** | **50.4** | **47.8** | **64.4** | **52.0** |

Results for competing methods are taken from Fu et al. (2024), with the best highlighted in **bold**.

Table 8: **Results on COCO-$20^i$ (Kang et al., 2019; Lin et al., 2014).** We report nAP (IoU thresholds 0.5–0.95), nAP50 (IoU 0.5), and nAP75 (IoU threshold 0.75) on novel classes.

**CD-FSOD (Xiong, 2023).** The Cross-Domain Few-Shot Object Detection (CD-FSOD) benchmark (Xiong, 2023) evaluates performance across six diverse scene domains, including photorealistic, cartoon, aerial, underwater, and industrial settings. This benchmark is particularly challenging, as it tests not only the few-shot learning capability but also cross-domain generalization. Unlike Pascal or COCO, there are no base categories in CD-FSOD, and only few-shot samples are provided for each domain. Following (Espinosa et al., 2025), we report nAP as the evaluation metric.

## A.4 FEATURE EXTRACTOR

We provide performance comparisons of DINOv2-B, DINOv2-L and a new model RADIOv4 (Ranzinger et al., 2026) on Pascal-$5^i$, COCO-$20^i$ and CD-FSOD datasets. From the results in Table 7, Table 8, and Table 9, we find that FSOD-VFM-RADIOv4 outperforms FSOD-VFM-DINOv2-B and FSOD-VFM-DINOv2-L in most of the sub-sets. Consequently, RADIOv4 is a better feature extractor than DINOv2 for our main experiments.

| Method | F.T Novel. | ArTaxOr | Clip art1k | DIOR | Deep Fish | NEU DET | UODD | Avg |
|---|---|---|---|---|---|---|---|---|
| TFA w/cos (Wang et al., 2020)° | ✓ | 3.1 / 8.8 / 14.8 | - | 8.0 / 18.1 / 20.5 | - | - | 4.4 / 8.7 / 11.8 | - |
| FSCE ○ (Sun et al., 2021)° | ✓ | 3.7 / 10.2 / 15.9 | - | 8.6 / 18.7 / 21.9 | - | - | 3.9 / 9.6 / 12.0 | - |
| DeFRCN (Qiao et al., 2021)° | ✓ | 3.6 / 9.9 / 15.5 | - | 9.3 / 18.9 / 22.9 | - | - | 4.5 / 9.9 / 12.1 | - |
| Distill-cdfsd (Xiong, 2023)° | ✓ | 5.1 / 12.5 / 18.1 | 7.6 / 23.3 / 27.3 | 10.5 / 19.1 / 26.5 | - / 15.5 / 15.5 | - / 16.0 / 21.1 | 5.9 / 12.2 / 14.5 | - / 16.4 / 20.5 |
| ViTDeT-FT† (Li et al., 2022)† | ✓ | 5.9 / 20.9 / 23.4 | 6.1 / 23.3 / 25.6 | 12.9 / 23.3 / 29.4 | 0.9 / 9.0 / 6.5 | 2.4 / 13.5 / 15.8 | 4.0 / 11.1 / 15.6 | 5.4 / 16.9 / 19.4 |
| Detic-FT (Zhou et al., 2022)† | ✓ | 3.2 / 8.7 / 12.0 | 15.1 / 20.2 / 22.3 | 4.1 / 12.1 / 15.4 | 9.0 / 14.3 / 17.9 | 3.8 / 14.1 / 16.8 | 4.2 / 10.4 / 14.4 | 6.6 / 13.3 / 16.5 |
| DE-ViT-FT (Zhang et al., 2023)† | ✓ | 10.5 / 38.0 / 49.2 | 13.0 / 38.1/40.8 | 14.7 / 23.4 / 25.6 | 19.3 / 21.2 / 21.3 | 0.6 / 7.8 / 8.8 | 2.4 / 5.0 / 5.4 | 10.1 / 22.3 / 25.2 |
| CD-ViTO (Fu et al., 2024)† | ✓ | 21.0 / 47.9 / 60.5 | 17.7 / 41.1 / 44.3 | 17.8 / 26.9 / 30.8 | 20.3 / 22.3 / 22.3 | 3.6 / 11.4 / 12.8 | 3.1 / 6.8 / 7.0 | 13.9 / 26.1 / 29.6 |
| Mixture (Liu et al., 2025) | ✓ | 26.1 / 63.3 / **71.3** | 20.1 / 45.1 / 49.9 | **20.6 / 32.1 / 37.8** | 24.2 / 29.5 / 34.1 | **9.1/ 19.0 / 23.7** | 9.0 / 19.6 / **22.1** | 18.2 / **34.7 / 39.8** |
| Meta-RCNN (Yan et al., 2019)° | ✗ | 2.8 / 8.5 / 14.0 | - | 7.8 / 17.7 / 20.6 | - | - | 3.6 / 8.8 / 11.2 | - |
| Detic (Zhou et al., 2022)† | ✗ | 0.6 / 0.6 / 0.6 | 11.4 / 11.4 / 11.4 | 0.1 / 0.1 / 0.1 | 0.9 / 0.9 / 0.9 | 0.0 / 0.0 / 0.0 | 0.0 / 0.0 /0.0 | 2.2 / 2.2 / 2.2 |
| DE-ViT (Zhang et al., 2023)† | ✗ | 0.4 / 10.1 / 9.2 | 0.5 / 5.5 / 11.0 | 2.7 / 7.8 / 8.4 | 0.4 / 2.5 / 2.1 | 0.4 / 1.5 / 1.8 | 1.5 / 3.1 /3 .1 | 1.0 / 5.1 / 5.9 |
| No-Time-To-Train (Espinosa et al., 2025) | ✗ | 28.2 / 35.7 / 35.0 | 18.9 / 24.9 / 25.9 | 14.9 / 18.5 / 16.4 | 30.5 / 28.9 / 29.6 | 5.5 / 5.2 / 5.5 | 10.0/ **20.2** / 16.0 | 18.0 / 22.4 / 21.4 |
| **FSOD-VFM-DINOv2-B** | ✗ | 50.6 / 55.6 / 57.3 | 25.1 / 40.0 / 41.6 | 16.0 / 20.0 / 19.8 | 34.1 / 33.0 / 33.3 | 6.1 / 5.7 / 5.8 | 9.1 / 16.0 / 17.3 | 23.5 / 28.4 / 29.2 |
| **FSOD-VFM-DINOv2-L** | ✗ | **51.4** / 62.0 / 61.5 | 29.1 / 43.7 / 46.5 | 18.3 / 23.5 / 22.5 | **35.0 / 33.9 / 34.3** | 5.9 / 7.4 / 7.2 | **11.8** / 17.3 / 17.5 | 25.3 / 31.3 / 31.6 |
| **FSOD-VFM-RADIOv4** | ✗ | 51.4 / **65.9** / 66.1 | **33.8 / 50.5 / 52.1** | 18.8 / 25.6 / 25.7 | 34.1/ 32.5 / 33.1 | 6.3 / 10.3 / 10.6 | 9.2 / 18.0 / 20.7 | **25.6** / 33.8 / 34.7 |

○ indicates Distill-CD-FSOD (Xiong, 2023) results, and † denotes CD-ViTO (Fu et al., 2024) results. Best results are in **bold**.

Table 9: **Results on the CD-FSOD benchmark (Xiong, 2023).** We report nAP (IoU thresholds 0.5–0.95) for all datasets, with each entry showing results for 1-shot, 5-shot, and 10-shot.

## A.5 Computational Overheads

We provide a summary of the number of parameters, FLOPs, and computation time for each model used, as shown in Table 10. Quantitative results reveal that SAM2 incurs a longer inference time than both UPN and DINOv2.

| Model | Parameters (M) | FLOPs (G) | Average Inference Time (s) |
|---|---|---|---|
| UPN (Jiang et al., 2024b) | 218 | 1523 | 0.33 |
| SAM2 (Ravi et al., 2024) | 225 | 844 | 0.86 |
| DINOv2 (Oquab et al., 2023) | 304 | 415 | 0.24 |

Table 10: **Computational Overheads of different components.** The inference device is A40 GPU.

## A.6 Component Ablations

We provide component ablation studies of our two main components: UPN and SAM2. Firstly, we conduct experiments without SAM2 on Pascal-5i (split 1, 1-shot) and COCO-5i (10-shot). In this case, we directly apply average pooling over the raw bounding box regions to obtain features. The results are shown in Table 11. The substantial performance drop demonstrates that SAM2 is essential for our method. Although SAM2 does not always produce perfect masks, having a reasonably accurate foreground mask is far more effective than using the entire bounding box for feature extraction. This validates the necessity of incorporating SAM2 in our pipeline.

We additionally conduct experiments without UPN, where SAM2 is used directly for object proposal and mask extraction. The results show a clear performance drop compared with using our full set of proposed modules.

| Settings | Pascal-5$^i$ | COCO-20$^i$ |
|---|---|---|
| Ours | 77.5 | 59.4 |
| w/o SAM2 (Ravi et al., 2024) | 25.5 | 15.5 |
| w/o UPN (Jiang et al., 2024b) | 56.7 | 41.5 |

Table 11: **Component ablation studies.** We report nAP50 (IoU 0.5) on novel classes.

## A.7 The Use of Large Language Models (LLMs)

Large Language Models (LLMs) were utilized to polish the writing in this manuscript.

