# OpenReview forum: "FSOD-VFM: Few-Shot Object Detection with Vision Foundation Models and Graph Diffusion"
_ICLR.cc/2026/Conference — ICLR 2026 Poster_

### Official Review · Reviewer_MxA8 · 2025-10-25

**Soundness:** 2
**Presentation:** 2
**Contribution:** 2
**Rating:** 2
**Confidence:** 5

**Summary:**

The paper addresses few-shot object detection (FSOD). It particularly focuses on training-free detectors by integrating multiple vision foundation models. It uses a universal proposal network (UPN) to detect object proposals and then SAM2 to output object masks. It uses DINOv2 to represent masked objects for matching test-time proposal and few-shot annotated images, achieving FSOD results. It evaluates on traditional benchmarks built on COCO, PASCAL and a few others. It compares against previous FSOD methods which train a base model (on significantly smaller set than foundation models' pretraining set) and finetune on few-shot novel classes. It reports superior performance than the previous FSOD methods.

**Strengths:**

Below are notable strengths of this paper.
- The use of graph difussion to post-process predictions based on confidence scores is interesting.
- Evaluation metrics are standard.

**Weaknesses:**

Below are major weaknesses of this paper.
- While the paper advocates training-free methods for FSOD, it is unclear how such training-free methods can detect objects specified by downstream tasks that require to detect non-trivial objects. For examples, if a downstream task requires amodal detection but SAM cannot output amodal segments, how training-free methods handle this discrepancy? Moreover, if a downstream task requires to detect rare objects, can the training-free detector handle this requirement?. The datasets used in experiments contain common objects (from COCO and PASCAL), which are likely learned by vision foundation models and hence disguise limitations of training-free FSOD detectors.

- Images in Figure 1 are low-quality. It seems that (b) and (c) contain "motorcycle" predictions and some more on the cows. There are also dim bounding boxes, likely post-processed in some way. Can authors clarify?

- From Figure 1, it is unclear how graph diffusion helps. From (d), (e) and (f), we can see that, even after graph diffusion which diffuses low confidence scores on x-axis, one can still set the confidence threshold as 0.5. So, it is not easy to follow what messages this figure mean to deliver.

- The paper partially follows a traditional FSOD protocol which evaluates on "novel classes" (Line363), but it does not exploit any "base classes" (Line368). This is a good practice given that it leverages foundation models which can be thought of having been trained on massive "base data". However, it is unfair to compare the proposed method with previous ones which use a much smaller "base data" set to pretrain models. For fair comparison, it should run existing FSOD algorithms on foundational detectors (e.g., DINO and GroundingDINO), which are assumed to be pretrained on the same "base data".

- The datasets used in this work are outdated and small-scale. First, Pascal and COCO were published in 2010 and 2014; but do vision foundation models see similar data to them in the pretraining dataset? Second, Pascal-5 and COCO-20 seems to mean the evaluation is on 5 and 20 classes, respectively. This is rather a small-scale and artificial setting.

- Hyperparameter setting is a big concern in FSOD. The core challenge in FSOD is the lack of data: limited training data for the few-shot (novel) classes and realistically no validation set. Therefore, when reporting the numeric metrics, do the authors tune hyperparameters on the test sets? In Section 4.2, Table 4 shows that the hyperparameter lambda has a significant effect on the final performance.

- In terms of the use of graph diffusion, the paper lacks intuition or rationale how to build a graph. The paper treats each detection as a node, but how to assign weights to edges? Do edges mean similarities between two detections? More important details and discussions are missing.

**Questions:**

The reviewer asks the authors to address each point in weaknesses listed above and does not repeat them in this Questions box.

---

> ### Author Response · Authors · 2025-11-22
> **Response (1/3) to Review MxA8, we remain open to further discussion.**
>
> **We greatly appreciate the reviewer’s insightful comments and offer a detailed response below. We also remain receptive to addressing any additional questions or further feedback you may have.**
>
> 1. > While the paper advocates training-free methods for FSOD, it is unclear how such training-free methods can detect objects specified by downstream tasks that require to detect non-trivial objects. For examples, if a downstream task requires amodal detection but SAM2 cannot output amodal segments, how training-free methods handle this discrepancy? Moreover, if a downstream task requires to detect rare objects, can the training-free detector handle this requirement?. The datasets used in experiments contain common objects (from COCO and PASCAL), which are likely learned by vision foundation models and hence disguise limitations of training-free FSOD detectors.
>
> **Response**: We fully agree that downstream tasks such as amodal detection and rare object detection present significant challenges to any training-free approach. **Our goal in this work is not to claim that training-free methods can solve all such tasks, but to demonstrate that recent advances in vision foundation models (VFMs) have substantially expanded the generalization capabilities of training-free pipelines, making them far more practical than before**. This enables us to leverage the most advanced vision foundation models to directly build an object detector across various domains, without any training.
>
> To evaluate generalization beyond common categories, we conducted experiments on the CD-FSOD benchmark, which spans six diverse and challenging domains (see Figure 3 and Figure 6), including photorealistic scenes, cartoons, aerial imagery, underwater scenes, and industrial defects. Our VFM was not adapted or fine-tuned for these rare domains. Yet, as shown in Table 3, our training-free FSOD-VFM outperforms several state-of-the-art training-based methods, indicating that VFMs can already provide strong, transferable object-level priors even in unfamiliar settings.
>
> These results highlight the key message of our work: **The rapid progress of VFMs enables a promising direction for FSOD, one that minimizes task-specific training and directly leverages the broad generalization already learned by foundation models.** This direction is highly relevant for real-world applications where data collection and retraining are costly, and we believe it deserves more attention from the community.
>
>
>
> 2. >  Images in Figure 1 are low-quality. It seems that (b) and (c) contain "motorcycle" predictions and some more on the cows. There are also dim bounding boxes, likely post-processed in some way. Can authors clarify?
>
> **Response**: We appreciate the reviewer’s careful examination of Figure 1. The “motorcycle” prediction observed in the figure is a prediction with low confidence score 0.02 and can be trivially removed in practice using a standard confidence threshold.
>
> Regarding the appearance of bounding boxes, we follow a clear visualization convention described in lines 101–103: darker boxes indicate higher confidence, while more transparent boxes indicate lower confidence. From subfigure b to c, our graph diffusion algorithm substantially suppresses redundant cow detections, which is reflected by the visibly lighter bounding boxes. This transparency change provides an intuitive visual illustration of how our method reduces redundant proposals by lowering their confidence scores.
>
> 3. >  From Figure 1, it is unclear how graph diffusion helps. From (d), (e) and (f), we can see that, even after graph diffusion which diffuses low confidence scores on x-axis, one can still set the confidence threshold as 0.5. So, it is not easy to follow what messages this figure mean to deliver.
>
> **Response**: We thank the reviewer for the insightful feedback. The purpose of Figure 1(d–f) is not to suggest that a fixed threshold (e.g., 0.5) suddenly becomes optimal after graph diffusion, but rather to illustrate the relative redistribution of confidence scores.
>
> Specifically, across Figures 1(d–f), high-quality proposals (IoU > 0.75 with any ground truth) consistently maintain high confidence, whereas low-quality proposals (IoU < 0.1) are substantially suppressed after graph diffusion. This widening separation between high- and low-quality boxes makes the confidence scores more semantically meaningful and greatly reduces sensitivity to threshold selection.
>
> This effect is also validated empirically: on Pascal-5i, COCO-20i, and CD-FSOD, the refined confidence distribution produced by graph diffusion leads to consistent and significant performance gains over prior methods, confirming that the mechanism meaningfully improves proposal quality rather than merely shifting absolute scores.

---

> ### Author Response · Authors · 2025-11-22
> **Response (2/3) to Review MxA8, we remain open to further discussion.**
>
> 4. > The paper partially follows a traditional FSOD protocol which evaluates on "novel classes" (line 363), but it does not exploit any "base classes" (line 368). This is a good practice given that it leverages foundation models which can be thought of having been trained on massive "base data". However, it is unfair to compare the proposed method with previous ones which use a much smaller "base data" set to pretrain models. For fair comparison, it should run existing FSOD algorithms on foundational detectors (e.g., DINO and GroundingDINO), which are assumed to be pretrained on the same "base data".
>
> **Response**: We appreciate the reviewer’s concern. Importantly, recent work has already begun exploring FSOD on top of foundational detectors. A representative state-of-the-art baseline, Mixture of DETR and Vision Foundation Models [1]: evaluates cross-domain FSOD using DINO Detector [2] and DINOv2 [3], both pretrained on large-scale data and fine-tuning on target datasets. The results are presented in the table below.
>
> | Setting                  | ArTaxOr       | clipart1k     | DIOR          | DeepFish      | NEU-DET       | UODD          | Avg.       |
> |--------------------------|---------------|---------------|---------------|---------------|---------------|---------------|---------------|
> | Mixture [1] (Finetune)   | 26.1 / 63.3 / 71.3 | 20.1 / 45.1 / 49.9 | 20.6 / 32.1 / 37.8 | 24.2 / 29.5 / 34.1 | 9.1 / 19.0 / 23.7  | 9.0 / 19.6 / 22.1  | 18.2 / 34.7 / 39.8 |
> | FSOD-VFM (Training-Free) | 51.4 / 62.0 / 61.5 | 29.1 / 42.2 / 46.5 | 18.3 / 23.5 / 22.5 | 35.0 / 33.9 / 34.3 | 5.9 / 7.2 / 7.2    | 11.8 / 20.5 / 17.5 | 25.3 / 31.3 / 31.6 |
>
> Note that our FSOD-VFM framework is purely training-free: it directly leverages the generalization capability of vision foundation models without any finetuning or domain-specific adaptation. Despite this significant handicap, FSOD-VFM still achieves substantially better 1-shot performance. This highlights the strength of our approach in low-shot scenarios.
>
> Beyond the numerical results, we would also like to emphasize an important qualitative observation:
>
> - For domains that are extremely rare or visually distant from common imagery, such as industrial defect detection (NEU-DET), vision foundation models naturally struggle when used directly without finetuning. In these settings, training-based approaches provide substantial gains, which we openly acknowledge in our “Limitations” section.
>
> - However, for domains closer to photorealistic imagery, such as ArTaxOr, clipart1k, or DeepFish, our training-free method achieves strong, and often superior, 1-shot performance, even compared to methods finetuned on the target domain.
>
> This trend underscores our central message:
>
> **Modern vision foundation models possess strong generalization ability, and by directly leveraging these models, training-free FSOD can achieve surprisingly competitive performance across many real-world domains.** At the same time, visually divergent or industrial domains may still benefit from domain-specific finetuning, an insight consistent with practical deployment considerations.
>
> We think these findings collectively highlight the value and practicality of exploring training-free FSOD, especially for real-world scenarios where annotation or task-specific pretraining is costly or infeasible.
>
> [1] Liu, Chang-Han, et al. "DON’T NEED RETRAINING: A Mixture of DETR and Vision Foundation Models for Cross-Domain Few-Shot Object Detection." NeurIPS, 2025.
>
> [2] Hao Zhang et al.  Dino: Detr with improved denoising anchor boxes for end-to-end object detection. ICLR, 2023.
>
> [3] Oquab, Maxime, et al. "DINOv2: Learning Robust Visual Features without Supervision." TMLR 2024
>
> 5. > The datasets used in this work are outdated and small-scale. First, Pascal and COCO were published in 2010 and 2014; but do vision foundation models see similar data to them in the pretraining dataset? Second, Pascal-5 and COCO-20 seems to mean the evaluation is on 5 and 20 classes, respectively. This is rather a small-scale and artificial setting.
>
> **Response**: It is true that Pascal and COCO are older datasets; however, they remain standard benchmarks in the few-shot object detection (FSOD) community, and their widely-used splits (Pascal-5i, COCO-20i) allow for fair and reproducible comparison with prior works (lines 359 - 369, appendix lines 794 - 809). As for pretraining dataset of used vision foundation model, we clarified in the 1st question of reviewer 6RnX.
>
> To further address concerns about scale and diversity, we also evaluate our FSOD-VFM framework on CD-FSOD, a more challenging and cross-domain benchmark that covers multiple rare and diverse domains, including photorealistic, cartoon, aerial, underwater, and industrial defect images. These experiments confirm that our approach generalizes well beyond small-scale, commonly-used datasets.

---

> > ### Author Response · Authors · 2025-11-22
> > **Response (3/3) to Review MxA8, we remain open to further discussion.**
> >
> > 6. > Hyperparameter setting is a big concern in FSOD. The core challenge in FSOD is the lack of data: limited training data for the few-shot (novel) classes and realistically no validation set. Therefore, when reporting the numeric metrics, do the authors tune hyperparameters on the test sets? In Section 4.2, Table 4 shows that the hyperparameter lambda has a significant effect on the final performance.
> >
> > **Response**: As noted in line 322, we use the same set of hyperparameters across all datasets (different dataset, different splits) and shot settings (1-shot, 5-shot and 10-shot), without any dataset-specific tuning. This ensures a fair and consistent evaluation.
> >
> > Moreover, Table 4 demonstrates that our method is not sensitive to the hyperparameter, performance remains strong across a wide range of values. These results confirm that our approach is robust to hyperparameter selection and that its effectiveness does not rely on careful per-dataset tuning.
> >
> > 7. > In terms of the use of graph diffusion, the paper lacks intuition or rationale how to build a graph. The paper treats each detection as a node, but how to assign weights to edges? Do edges mean similarities between two detections? More important details and discussions are missing.
> >
> > **Response**: The construction of the graph and the definition of edge weights are detailed in the manuscript. Specifically, as described in Equation 3 (lines 264–267), edge weights quantify the similarity between nodes using a mask-based intersection-over-self metric, and the node weights reflect the strength of the strongest outgoing edge (line 292). The subsequent update of node weights via graph diffusion is described in Equation 5 (line 299). These components together define how energy flows between nodes and how low-confidence nodes are suppressed while high-confidence nodes are preserved.

---

### Official Review · Reviewer_N1W4 · 2025-10-31

**Soundness:** 2
**Presentation:** 2
**Contribution:** 2
**Rating:** 4
**Confidence:** 4

**Summary:**

This paper proposes FSOD-VFM, a training-free few-shot object detection method that integrates three vision foundation models (UPN for proposal generation, SAM2 for mask extraction, and DINOv2 for feature extraction) to detect novel categories using only a few annotated samples. Its core contribution is a graph diffusion-based confidence reweighting mechanism that effectively mitigates proposal over-fragmentation and enhances detection granularity. Extensive experiments on Pascal-5, COCO-20 , and CD-FSOD benchmarks demonstrate that the method significantly outperforms existing training-free approaches and even surpasses many training-based methods, offering a novel perspective for solving visual tasks in a training-free paradigm.

**Strengths:**

1.  The paper introduces a graph diffusion confidence reweighting mechanism to address the prevalent issue of proposal fragmentation in training-free few-shot detection. Constructing a directed graph among proposals and simulating energy propagation to effectively distinguish between complete object proposals and partial fragment proposals, thereby significantly enhancing detection granularity.

2.  It is commendable that the authors have validated the method's effectiveness across multiple benchmarks (Pascal-5, COCO-20). The notable performance improvement on the cross-domain few-shot detection benchmark (CD-FSOD) particularly underscores the method's strong generalizability and universality, demonstrating its potential to tackle real-world, complex scenarios.

**Weaknesses:**

1.  The model is overly complex, employing three powerful foundation models to address a relatively specific issue (fragmented bounding boxes generated by UPN). The model complexity and computational cost are exceptionally high. SAM2 is a powerful segmentation model whose generated precise masks already contain rich structural information of objects. Why can't detection boxes be generated directly from SAM2's masks, or why can't a more streamlined pipeline be built around SAM2 as the core, rather than relegating it to an auxiliary role that merely generates masks for UPN proposals?

2.  The methodological description in the paper, particularly concerning the core Figure 3, is insufficient for readers to understand the entire technical pipeline. The limited informational content of the figure creates excessive jumps between key steps like "Feature Extraction & Matching" and "Graph Diffusion," making comprehension very difficult. To enhance the work's comprehensibility and reproducibility, it is strongly recommended to supplement the paper with more detailed flowcharts or schematic diagrams.

3.  The definition and explanation of the core concept of "Heat" in the graph diffusion process within the paper are highly ambiguous,easily causing significant confusion for readers. As described in Section 3.2, "Heat" flows from low-scoring nodes to high-scoring nodes, yet the final stable energy distribution (π̂) is used to penalize nodes, which seems counterintuitive. The authors must add a clear explanation explicitly distinguishing between the "process of energy flow" and the "meaning represented by the steady-state energy distribution."

4.  The paper fails to clearly differentiate the proposed "Graph Diffusion" method from the classical NMS mechanism, which severely undermines the perceived innovation of the method. Superficially, both aim to eliminate redundant boxes, but the authors do not delve into explaining their fundamental differences. The authors must include a comparative discussion in the text contrasting the underlying principles of their method with those of NMS.

5.  Regarding Table 3 and the CD-FSOD experimental results: the original benchmark's reported results are based on mAP (mean Average Precision), but the comparative results presented in this paper for CD-FSOD use nAP-50. This constitutes a serious methodological error in evaluation.

**Questions:**

Could the functionality of graph diffusion be achieved through other forms of NMS, based on a comparison of their underlying principles?

---

> ### Author Response · Authors · 2025-11-22
> **Response (1/2) to Review N1W4, we remain open to further discussion.**
>
> **We appreciate the reviewer’s insightful suggestion and provide our detailed response below. We remain open to addressing any further questions.**
>
> 1. > Question: The model is overly complex, employing three powerful foundation models to address a relatively specific issue (fragmented bounding boxes generated by UPN). The model complexity and computational cost are exceptionally high. SAM2 is a powerful segmentation model whose generated precise masks already contain rich structural information of objects. Why can't detection boxes be generated directly from SAM2's masks, or why can't a more streamlined pipeline be built around SAM2 as the core, rather than relegating it to an auxiliary role that merely generates masks for UPN proposals?
>
> **Response**: As suggested, we conduct the experiment without UPN, where we directly use the masks generated by SAM2. The performance comparison is shown below. For Pascal-5i, we report the AP50 averaged over the 1-shot performance of the three splits; for COCO-5i, we report the 10-shot AP50. As the results indicate, removing UPN leads to a substantial performance drop compared with our full FSOD-VFM model.
>
>
>
> | Settings  | Pascal-5i | COCO-5i   |
> |----------|:-----------|:--------|
> | Ours   | 77.5     | 59.4  |
> | w/o SAM2   | 25.5     | 15.5  |
> | w/o UPN   | 56.7     | 41.5  |
>
> The advantage of integrating UPN into our FSOD-VFM framework comes from two key factors. First, UPN generates high-quality bounding boxes. Second, UPN is trained to produce reliable confidence scores that accurately reflect bounding box quality: high-quality boxes receive higher scores, while low-quality boxes receive lower scores, as illustrated in Figure 1(d). Although UPN may occasionally produce over-fragmented proposals, this issue is effectively mitigated through the proposed graph diffusion.
>
> We also conduct experiments without SAM2, where proposal features are extracted using simple average pooling instead of applying RoI pooling over a meaningful foreground mask. This leads to a clear performance drop compared with our complete framework. We appreciate the reviewer’s suggestion, which motivated this ablation and made our paper more comprehensive.
>
> 2. > Question: The methodological description in the paper, particularly concerning the core Figure 3, is insufficient for readers to understand the entire technical pipeline. The limited informational content of the figure creates excessive jumps between key steps like "Feature Extraction & Matching" and "Graph Diffusion," making comprehension very difficult. To enhance the work's comprehensibility and reproducibility, it is strongly recommended to supplement the paper with more detailed flowcharts or schematic diagrams.
>
> **Response**: We appreciate the valuable suggestion. The revised figure has been added to the Appendix in the updated manuscript (Lines 922–938). We welcome any further feedback and will refine the main-paper version in the updated revision.
>
> 3. > The definition and explanation of the core concept of "Heat" in the graph diffusion process within the paper are highly ambiguous,easily causing significant confusion for readers. As described in Section 3.2, "Heat" flows from low-scoring nodes to high-scoring nodes, yet the final stable energy distribution (π̂) is used to penalize nodes, which seems counterintuitive. The authors must add a clear explanation explicitly distinguishing between the "process of energy flow" and the "meaning represented by the steady-state energy distribution."
>
> **Response**: Thank you for highlighting this ambiguity. We agree that our original description did not clearly distinguish the direction of the diffusion process from the interpretation of the steady-state energy distribution.
>
> To clarify, during the diffusion process, energy indeed flows from low-scoring nodes to high-scoring nodes. This models how reliable (high-score) proposals accumulate more “support” from their neighbors. However, for our task, what we ultimately need is a measure that penalizes unreliable (low-score) nodes. Therefore, after obtaining the steady-state distribution, we apply a simple transformation, which consists of taking its negative and adding 1, to convert it into a penalty score. This transformation ensures that nodes with low final energy receive stronger penalties, while high-energy nodes are preserved.
>
> We will revised the paper to clearly separate these two concepts and improved the explanation of both the diffusion dynamics and the meaning of the steady-state values according to our final discussion.

---

> > ### Author Response · Authors · 2025-11-22
> > **Response (2/2) to Review N1W4, we remain open to further discussion.**
> >
> > 4. >  The paper fails to clearly differentiate the proposed "Graph Diffusion" method from the classical NMS mechanism, which severely undermines the perceived innovation of the method. Superficially, both aim to eliminate redundant boxes, but the authors do not delve into explaining their fundamental differences. The authors must include a comparative discussion in the text contrasting the underlying principles of their method with those of NMS.
> >
> > **Response**: Thank you for the comment. We agree that the distinction between Graph Diffusion and NMS needs to be made clearer. Although both reduce redundant proposals, their underlying principles are fundamentally different.
> >
> > **NMS makes hard suppression decisions based solely on bounding box IoU and confidence scores.** If two boxes overlap beyond a fixed threshold, the lower-score box is directly removed. This binary rule often suppresses valid detections when objects overlap or when bounding boxes are imperfect.
> >
> > **Graph diffusion, in contrast, does not remove boxes based on thresholds.** Instead, it refines proposal scores by propagating information between proposals using mask-level relationships from SAM2, which capture object boundaries much more accurately than bounding boxes. Through diffusion, unreliable proposals naturally receive lower scores, while reliable ones gain consistent support.
> >
> > In short, NMS deletes boxes using IoU thresholds, whereas Graph Diffusion adjusts scores using mask-level similarity without any hard removal, making it more robust to occlusion and box misalignment.
> >
> > We will revise expression of Graph Diffusion in the next version according to further feedbacks.
> >
> > 5. >  Regarding Table 3 and the CD-FSOD experimental results: the original benchmark's reported results are based on mAP (mean Average Precision), but the comparative results presented in this paper for CD-FSOD use nAP-50. This constitutes a serious methodological error in evaluation.
> >
> > **Response**: Thank you for raising this concern. We would like to clarify that there is a misunderstanding regarding Table 3. All results reported in Table 3 are evaluated using mAP (i.e., the average AP from IoU 0.5 to 0.95), which is the same evaluation metric used in the original CD-FSOD benchmark. We do not use nAP-50 for these comparisons (c.f. lines 361 - 364, caption of Table 3, appendix lines 848 - 854).
> >
> > Therefore, our evaluation is fully aligned with prior work, and the improvements reported by FSOD-VFM are based on a fair and consistent metric.
> >
> > 6. >  Could the functionality of graph diffusion be achieved through other forms of NMS, based on a comparison of their underlying principles?
> >
> > **Response**: Thank you for the thoughtful question. We would like to clarify the relationship between Graph Diffusion and NMS-based approaches:
> >
> > - We already compare against strong NMS variants commonly used in object detection, including NMS, Soft-NMS and WBF (Table 4). These methods represent the advanced forms of NMS, and our proposed Graph Diffusion consistently outperforms them, demonstrating its effectiveness beyond the capabilities of NMS-style suppression.
> >
> > - As discussed in our response to Question 4, Graph Diffusion differs fundamentally from NMS. NMS applies hard or soft suppression based on box IoU, while Graph Diffusion performs confidence refinement using mask-level relationships from SAM2. This allows our method to handle occlusion, misalignment, and part-level proposals more robustly than IoU-based suppression.
> >
> > - Regarding whether other NMS variants could replicate Graph Diffusion: We are open to further exploration. We welcome suggestions of specific NMS variants for further evaluation. However, based on our current experiments with strong NMS baselines, we have not found an NMS-style method that can fully replace the benefits brought by Graph Diffusion.

---

### Official Review · Reviewer_JBSW · 2025-10-31

**Soundness:** 2
**Presentation:** 2
**Contribution:** 2
**Rating:** 4
**Confidence:** 4

**Summary:**

The paper proposes the FSOD-VFM framework for few-shot object detection by combining several powerful vision foundation models (such as DINOv2, SAM2). It proposes a graph diffusion module that treats bounding box proposals as nodes in a graph and uses graph diffusion operations to adjust the confidence of each bounding box proposals, effectively suppressing local, fragmented boxes. The method performance across multiple benchmark datasets. Experimental study on cross domain datasets show improved performance over the counterpart methods.

**Strengths:**

1. The proposed graph diffusion method effectively suppresses the overfragmentation without training.

2. The proposed method is simple and easy to follow.

3. The writing of the paper is clear and fluent.

**Weaknesses:**

There are issues with method:

a) The graph diffusion module uses Equation 3 to calculate the diffusion energy between two proposed bounding boxes. However, Equation 3 relies heavily on the confidence scores provided by UPN. If fragmented bounding box have higher confidence scores, the graph diffusion process can not improve the confidence of high-quality candidate boxes.

b) The method in the paper uses SAM to segment the bounding box proposals given by UPN, in order to obtain cleaner foreground regions. However, the paper does not explain whether the method's performance would be significantly degrade if SAM fails to generate accurate masks. It would be helpful if the authors could visualize the masks provided by SAM in the cross domain tasks and conduct a comparison experiment without using SAM , to validate the necessity of incorporating SAM into the method.

**Questions:**

In addition to the questions raised in the Weaknesses section it would be great to hear authors' thoughst on why the performance in the 10-shot setting is lower than in the 5-shot setting or 1-shot setting in the table 3.

---

> ### Author Response · Authors · 2025-11-22
> **Response (1/2) to Review JBSW, we remain open to further discussion.**
>
> **We appreciate the reviewer’s insightful suggestion and provide our detailed response below. We remain open to addressing any further questions.**
>
> 1. > Question: The graph diffusion module uses Equation 3 to calculate the diffusion energy between two proposed bounding boxes. However, Equation 3 relies heavily on the confidence scores provided by UPN. If fragmented bounding box have higher confidence scores, the graph diffusion process can not improve the confidence of high-quality candidate boxes.
>
> **Response**: We acknowledge that, according to Equation 3, a fragmented bounding box (low-quality proposal) with a higher confidence score than a complete box would be a challenging case. However, this is not unique to our method; standard post-processing strategies, such as NMS, Soft-NMS [1], and WBF [2], similarly rely on confidence scores to reflect proposal quality. Importantly, the edge connections in Equation 3 still mitigate this concern: fragmented boxes are penalized more heavily due to the smaller denominator, while complete boxes experience a milder effect.
>
> UPN is trained as a class-agnostic object detector (CAOD), with confidence scores explicitly optimized to correlate with the overlap of a proposal with the underlying object. As a result, UPN naturally assigns higher scores to complete-object boxes and lower scores to fragmented or low-quality ones. This property is a key reason why CAOD is widely adopted in few-shot object detection, e.g., DeViT [3].
>
> In practice, CAOD ensures that high-quality proposals _generally dominate_ low-quality ones. This trend is illustrated in Figures 1(d), 1(e), and 1(f), where high-quality boxes (IoU > 0.75) typically have higher scores, while low-quality boxes are scored lower. Nevertheless, overlapping low-quality boxes can still affect ranking and overall performance. After applying our graph diffusion module, the scores of low-quality boxes are further reduced, resulting in a clearer separation between high-quality and low-quality proposals and improving overall detection quality.
>
> [1] Bodla, Navaneeth, et al. "Soft-NMS--improving object detection with one line of code." ICCV. 2017.
> [2] Solovyev, Roman, Weimin Wang, and Tatiana Gabruseva. "Weighted boxes fusion: Ensembling boxes from different object detection models." Image and Vision Computing, 2021.
> [3] Zhang, Xinyu, et al. "Detect everything with few examples." CoRL 2024.
>
>
> 2. > Question: The method in the paper uses SAM to segment the bounding box proposals given by UPN, in order to obtain cleaner foreground regions. However, the paper does not explain whether the method's performance would be significantly degrade if SAM fails to generate accurate masks. It would be helpful if the authors could visualize the masks provided by SAM in the cross domain tasks and conduct a comparison experiment without using SAM , to validate the necessity of incorporating SAM into the method.
>
> **Response**: Thanks for the suggestion. In our design, obtaining a clean foreground mask is crucial for extracting high-quality features for matching. To assess the necessity of SAM2, we conducted experiments on Pascal-5i (split 1, 1-shot) and COCO-5i (10-shot) without using SAM2. In this case, we directly apply average pooling over the raw bounding box regions to obtain features. The results are shown below:
>
>
> | Settings  | Pascal-5i | COCO-5i   |
> |----------|:-----------|:--------|
> | Ours   | 77.5     | 59.4  |
> | w/o SAM2   | 25.5     | 15.5  |
> | w/o UPN   | 56.7     | 41.5  |
>
> The substantial performance drop demonstrates that SAM2 is essential for our method. Although SAM2 does not always produce perfect masks, as shown in the updated supplementary material (lines 948-968), having a reasonably accurate foreground mask is far more effective than using the entire bounding box for feature extraction. This validates the necessity of incorporating SAM2 in our pipeline.
>
> We additionally conduct experiments without UPN, where SAM2 is used directly for object proposal and mask extraction. The results show a clear performance drop compared with using our full set of proposed modules. We appreciate the suggestion, which motivated this ablation study and helped make the paper more complete.

---

> ### Author Response · Authors · 2025-11-22
> **Response (2/2) to Review JBSW, we remain open to further discussion.**
>
> 3. > Question: In addition to the questions raised in the Weaknesses section it would be great to hear authors' thoughst on why the performance in the 10-shot setting is lower than in the 5-shot setting or 1-shot setting in the table 3.
>
> **Response**: We appreciate the reviewer’s observation. This is indeed an interesting phenomenon: although 10-shot settings generally improve performance, certain classes exhibit slight degradation compared to 5-shot or even 1-shot results.
>
> This behavior stems from a core limitation of training-free few-shot methods. In our framework, support samples are not used for learning; instead, they are directly aggregated to form class-wise feature prototypes. Consequently, providing more samples (e.g., 10-shot) does not guarantee a better class representation as the additional samples may introduce noise, background clutter, or distribution mismatch. In such cases, the aggregated 10-shot prototype may deviate further from the true test-time distribution than the 5-shot prototype.
>
> This effect has also been reported in related works, such as No-Time-To-Train [1] and DE-ViT [2], which observe similar non-monotonic trends when increasing the number of shots in training-free settings. This is also mentioned in our limitation paragraph (c.f. lines 466 - 472)
>
> Importantly, when averaging over all classes (Table 3), the overall 10-shot performance is still slightly higher than 5-shot, indicating that the benefits generally outweigh the noise, and randomness diminishes with larger sample sizes.
>
> This also reveals a valuable direction for improvement. For example, developing sample-selection or prototype-refinement strategies could mitigate noise and help ensure that additional shots consistently contribute useful information. However, such extensions are out of scope for the current work, we leave this line of exploration as promising future work.
>
> [1] Espinosa, Miguel, et al. "No time to train! Training-Free Reference-Based Instance Segmentation." arXiv:2507.02798, 2025.
>
> [2] Zhang, Xinyu, et al. "Detect everything with few examples." arXiv:2309.12969, 2023.

---

### Official Review · Reviewer_6RnX · 2025-11-02

**Soundness:** 4
**Presentation:** 4
**Contribution:** 4
**Rating:** 8
**Confidence:** 5

**Summary:**

The paper proposed a graph-diffusion based method for FSOD problem by integrating foundation models. The core novelty is to build a foundation model-based bounding box proposals module and a graph diffusion module.  The proposed network can achieve SOTA performance on classic FSOD dataset with significant improvement.

**Strengths:**

1. The idea of utilizing vision foundation models is novel.
2. Formulas are very helpful to understand the core idea of graph diffusion and the motivation of the research.
3. The proposed method also shows good generalization to cross-domain scenario, which is remarkable.
4. The proposed method achieved SOTA performance.
5. The limitation section provides good analysis of the proposed method, especially the part about the marginal improvement when the  annotated samples increases.

**Weaknesses:**

1. In the settings of FSOD and CD-FSOD, the basic assumption is that the novel category objects are either never seen in the training set or not labeled. However, with the vision foundation model, how do the authors make sure that the novel objects is absolutely unseen? This part of statement should be more clear.
2. The proposed network explores a very different way comparing to classic FSOD paradigms, what would be the computational overhead in  the proposed networks setting?

**Questions:**

See weaknesses

---

> ### Author Response · Authors · 2025-11-22
> **Response to Review 6RnX, we remain open to further discussion.**
>
> **We appreciate the reviewer’s insightful suggestion and provide our detailed response below. We remain open to addressing any further questions.**
>
> 1. > Question: In the settings of FSOD and CD-FSOD, the basic assumption is that the novel category objects are either never seen in the training set or not labeled. However, with the vision foundation model, how do the authors make sure that the novel objects is absolutely unseen? This part of statement should be more clear.
>
> **Response**: Thank you for raising this important question. Ensuring that novel categories remain unseen is indeed a key requirement in FSOD and CD-FSOD evaluation. To address this concern, we provide a detailed clarification of the data sources used for the foundation models employed in our work and why they do not introduce label leakage or unfair advantages.
>
> _First_, our backbone vision foundation models (DINOv2, SAM2, and UPN) are all trained under self-supervised or class-agnostic objectives, meaning that no category-level information from our evaluation datasets is injected into these models. This property fundamentally prevents any leakage of novel class labels.
> _Second_, we carefully examined the training data of each model to ensure that there is no overlap (label or image) with our evaluation benchmarks, except for a controlled and well-understood case with UPN. Specifically:
> - DINOv2 is trained on the LVD-142M curated dataset [1], which includes the Pascal VOC training split but has no image-level overlap with COCO or CD-FSOD. Importantly, because DINO-v2 is trained using a purely self-supervised objective, it does not encode any category semantics from Pascal VOC or any other benchmark, ensuring that no label or semantic information leaks into our evaluation.
> - SAM2 is trained on the SA-V dataset [2], which, according to its description, has no image overlap with our evaluation benchmarks. Moreover, SAM2 is optimized for mask generation and does not use any object category supervision. Therefore, even with large-scale pretraining, its capabilities do not depend on semantic label information from our test categories.
> - UPN, which we use solely for class-agnostic object proposal generation, is trained on the COCO [4] and Objects365 [5] training splits. Its exposure to COCO enhances only its ability to localize generic object regions, not to classify them. Importantly:
>   - In COCO evaluation, UPN may produce stronger localization due to domain familiarity, but it never learns or uses any category labels, ensuring that the novel-class labels remain unseen.
>   - For Pascal and CD-FSOD, UPN has no training exposure, so both object regions and class labels remain entirely unseen, guaranteeing fair evaluation.
>
> Overall, our pipeline strictly prevents novel-class leakage under FSOD/CD-FSOD assumptions. The evaluation remains fair because **all models are trained without category supervision**.
>
> We would greatly appreciate any further suggestions you may have. We will incorporate the clarifications from this discussion into the revised manuscript to strengthen the transparency of our experimental setup.
>
> References:
> - [1] Oquab, Maxime, et al. "DINOv2: Learning Robust Visual Features without Supervision." TMLR, 2024
> - [2] Ravi, Nikhila, et al. "SAM2: Segment Anything in Images and Videos." ICLR, 2025
> - [3] Jiang, Qing, et al. "Chatrex: Taming multimodal llm for joint perception and understanding." arXiv, 2024
> - [4] Tsung-Yi Lin, et al. "Microsoft coco: Common objects in context." ICCV, 2014
> - [5] Shuai Shao, et al. "Objects365:  A large-scale, high-quality dataset for object detection." ICCV, 2019
>
>
> 2. > Question: The proposed network explores a very different way comparing to classic FSOD paradigms, what would be the computational overhead in the proposed networks setting?
>
> **Response:** As suggested, we provide a summary of the number of parameters, FLOPs, and computation time for each model used, as shown below:
>
> | Model  | Param. (M) | FLOPs (G)   | Avg. Inf.(s)   |
> |----------|:-----------|:--------|:--------|
> | UPN      | 218     | 1523   |0.33|
> | SAM2     | 225     | 844  |0.86|
> | DINOv2   | 304     | 415  |0.24|
>
> The inference time for a single image on an NVIDIA A40 GPU is approximately 2.4 seconds (c.f. Table 5). Roughly half of this time is spent on model inference, while the remainder is used for graph construction and graph diffusion. Unlike typical deep neural network inference, where most computations are efficiently executed on CUDA, our current implementation is primarily in Python, with both cosine similarity computation and graph diffusion remaining unoptimized.

---

> > ### Comment · Reviewer_6RnX · 2025-11-22
> >
> > Thanks, the author's response has addressed my concerns.

---

### Author Response · Authors · 2025-12-03
**Overview of Rebuttal and Key Clarifications**

Dear AC and Reviewers, below is a structured meta-response to key concerns.


### 1 Fair Comparison with Foundation Models (Reviewers 6RnX, MxA8)

- **Novel Class Unseen Guarantee**: Our backbone models (DINOv2, SAM2, UPN) are trained under self-supervised or class-agnostic objectives, with no category-level information from our evaluation datasets (COCO, Pascal, CD-FSOD) injected during pre-training. We have verified no image/label overlap between model pre-training data and our benchmarks, ensuring novel classes remain truly unseen.

(For details, please refer to our responses to 6RnX Q1.)

- **Fair Baseline Alignment**: We have added comparisons with state-of-the-art finetuning-based FSOD methods built on foundational detectors [1]. Our training-free approach still achieves superior performance on 1-shot setting,  demonstrating the strength of our framework.

(For details, please refer to our responses to MxA8 Q4.)

### 2 Methodological Clarity (Reviewers N1W4, MxA8)

- **Graph Diffusion Mechanism**: We have supplemented detailed explanations of graph construction (node definition, edge weight calculation via mask-based similarity) and the "Heat" energy flow process.

(For details, please refer to our responses to N1W4 Q3,Q4, and MxA8 Q7.)

- **Distinction from NMS**: We explicitly contrast graph diffusion with NMS and its variants (Soft-NMS, WBF) in terms of underlying principles: NMS uses hard/soft suppression based on bounding box IoU, while graph diffusion refines confidence scores via mask-level relationships without hard removal.

(For details, please refer to our responses to N1W4 Q4 and Q6.)

### 3 Computational Complexity and Pipeline Necessity (Reviewers JBSW, N1W4, 6RnX)

- **Module Necessity**: Ablation experiments demonstrate that removing any core module (UPN, SAM2) leads to significant performance drops. SAM2’s role in extracting clean foreground masks is irreplaceable, while UPN provides high-quality object proposals with reliable confidence scores.

(For details, please refer to our responses to N1W4 Q1 and JBSW Q2.)

- **Computational Efficiency**: We have supplemented detailed efficiency metrics for each component.

(For details, please refer to our responses to 6RnX Q2.)

### 4 Generalization and Evaluation Rigor (Reviewers MxA8)

- **Cross-Domain Generalization**: We emphasize that our method is evaluated on CD-FSOD, a diverse benchmark covering six challenging domains (aerial, underwater, industrial defects, etc.). The superior performance of our method on it demonstrates its contribution.

(For details, please refer to our responses to MxA8 Q1.)

- **Hyperparameter Robustness**: We confirm that all experiments use the same set of hyperparameters across datasets and shot settings (1-shot/5-shot/10-shot) without per-dataset tuning.

(For details, please refer to our responses to MxA8 Q6.)

## References
[1] Liu, Chang-Han, et al. "DON’T NEED RETRAINING: A Mixture of DETR and Vision Foundation Models for Cross-Domain Few-Shot Object Detection." NeurIPS, 2025.

---

### Public Comment · ~Chen-Bin_Feng1 · 2026-03-02
**Camera Ready Revision**

Dear AC and Reviewers,
Thank you very much for your constructive comments and valuable suggestions. We sincerely appreciate the time and effort you dedicated to reviewing our work. In the camera-ready version, we have carefully incorporated your feedback and made the following modifications:
1. We release our code for reproduction: https://intellindust-ai-lab.github.io/projects/FSOD-VFM/
2. We have revised and updated the model overview figure to make it more comprehensive and illustrative (see Figure 2 on Page 4).
3. We improved the definition and explanation of the core concept of "Heat" (See Section 3.2, subsection: Graph Diffusion on Page 6).
4. We added a section to clarify the difference between Graph Diffusion and NMS (See Section 3.2, subsection: Comparison between Graph Diffusion and NMS on Page 6).
5. We added comparisons between a state-of-the-art training-based few-shot detection model, Mixture [1], in our CD-FSOD model performance comparison (See Table 3 on Page 8 and Table 9 in the supplementary on Page 17).
6. We added a segmentation visualization diagram to illustrate the segmentation results. (See Figure 4 on Page 9).
7. We added a computation time analysis (See Appendix section A.5 on Page 17).
8. We added an ablation study of different components in our model (See Appendix section A.6 on Page 17).
9. For completeness, we also conducted additional experiments using Radiov4 [2], a newly released vision foundation model, and integrated it into our framework. The corresponding results are reported in Appendix Tables 7–9 (Pages 16–17) and Appendix Section A.4 (Page 16).

[1] Liu, Chang-Han, et al. "DON’T NEED RETRAINING: A Mixture of DETR and Vision Foundation Models for Cross-Domain Few-Shot Object Detection." NeurIPS, 2025.

[2] Mike Ranzinger, Greg Heinrich, Collin McCarthy, Jan Kautz, Andrew Tao, Bryan Catanzaro, and Pavlo Molchanov. C-radiov4 (tech report). arXiv preprint arXiv:2601.17237, 2026.

---

### Meta-Review · Area_Chair_8juD · 2026-01-06

**Summary:**

The authors actually did a great job in their summary of they concerns raised by the reviewers:
1 Fair Comparison with Foundation Models (Reviewers 6RnX, MxA8)
2 Methodological Clarity (Reviewers N1W4, MxA8)
3 Computational Complexity and Pipeline Necessity (Reviewers JBSW, N1W4, 6RnX)
4 Generalization and Evaluation Rigor (Reviewers MxA8)

I feel that there was a robust rebuttal made for each of them and given their intial rating addressing the concerns meant that the ratings of the paper would have changed

**Reviewer Concerns:**

The ablation helped quantify the need for each component and this should address the issues raised by 6RnX ,JBSW, N1W4 and this probably was by far the single rebuttal evidence that cut across many reviewer comments.

Other issues that are addressed are
1. Whether novel classes remain truly unseen given VFM pretraining (6RnX, addressed)
2. Graph diffusion mechanism clarity and distinction from NMS (N1W4, addressed)

The issues that are partiall addressed are generalization to truly rare/amodal objects (acknowledged as limitation)

The issues that are still outstanding are Inherent limitation of training-free methods for visually distant domains

**Reviewer Scores:**

I feel the biggest movement would be for JBSW and N1W4 moving them to the to above the threshold territory since their concerns were addressed. Hence, the submission would have moved to a bit overall but not necessarily in the clearly accept category but still made a clear case of addressing the issues of the reviwers.

---

### Decision · Program_Chairs · 2026-01-26

Accept (Poster)